

# Use of Lidar Aerosol Extinction and Backscatter Coefficients to Estimate Cloud Condensation Nuclei (CCN) Concentrations in the Southeast Atlantic

Emily D. Lenhardt[1], Lan Gao[1], Jens Redemann[1], Feng Xu[1], Sharon P. Burton[2], Brian Cairns[3], Ian Chang[1], Richard A. Ferrare[2], Chris A. Hostetler[2], Pablo E. Saide[4,5], Calvin Howes[4], Yohei Shinozuka[6], Snorre Stamnes[2], Mary Kacarab[7], Amie Dobracki[8], Jenny Wong[9], Steffen Freitag[10], Athanasios Nenes[11,12]

[1]School of Meteorology, University of Oklahoma, Norman, OK, 73072, United States
[2]NASA Langley Research Center, Hampton, VA, 23666, United States
[3]NASA Goddard Institute for Space Studies, New York, NY, 10025, United States
[4]Department of Atmospheric and Oceanic Sciences, University of California – Los Angeles, Los Angeles, CA, 90095, United States
[5]Insitute of the Environment and Sustainability, University of California – Los Angeles, Los Angeles, CA, 90095, United States
[6]Bay Area Environmental Research Institute, Moffett Field, CA, 94035, United States
[7]School of Earth and Atmospheric Sciences, Georgia Institute of Technology, Atlanta, GA, 30332, United States
[8]Department of Atmospheric Sciences, University of Miami, Miami, FL, 33146, United States
[9]Department of Chemistry and Biochemistry, Mount Allison University, Sackville, New Brunswick, E4L 1E2, Canada
[10]State Agency for Nature, Environment and Consumer Protection North Rhine-Westphalia, Recklinghausen, 45659, Germany
[11]Institute for Chemical Engineering Sciences, Foundation for Research and Technology, Hellas, Patras, GR-26504, Greece
[12]School of Architecture, Civil & Environmental Engineering, Ecole Polytechnique fédérale de Lausanne, CH-1015, Lausanne, Switzerland

*Correspondence to*: Emily D. Lenhardt (emily.lenhardt@ou.edu)

**Abstract.** Accurately capturing cloud condensation nuclei (CCN) concentrations is key to understanding the aerosol-cloud interactions that continue to feature the highest uncertainty amongst numerous climate forcings. In situ CCN observations are sparse and most non-polarimetric passive remote sensing techniques are limited to providing column-effective CCN proxies such as total aerosol optical depth (AOD). Lidar measurements, on the other hand, resolve profiles of aerosol extinction and/or backscatter coefficients that are better suited for constraining vertically-resolved aerosol optical and microphysical properties. Here we present relationships between aerosol backscatter and extinction coefficients measured by the airborne High Spectral Resolution Lidar 2 (HSRL-2) and in situ measurements of CCN concentrations. The data were obtained during three deployments in the NASA ObseRvations of Aerosols above Clouds and their intEractionS (ORACLES) project, which took place over the Southeast Atlantic (SEA) during September 2016, August 2017, and September-October 2018.





Our analysis of spatiotemporally collocated in situ CCN concentrations and HSRL-2 measurements indicates strong linear relationships between both data sets. The correlation is strongest for supersaturations greater than 0.25% and dry ambient conditions above the stratocumulus deck, where relative humidity (RH) is less than 50%. We find CCN – HSRL-2 Pearson correlation coefficients between 0.95-0.97 for different parts of the seasonal burning cycle that suggest fundamental similarities in biomass burning aerosol (BBA) microphysical properties. We find that ORACLES campaign-average values of in situ CCN and in situ extinction coefficients are qualitatively similar to those from other regions and aerosol types, demonstrating overall representativeness of our data set. We compute CCN – backscatter and CCN – extinction regressions that can be used to resolve vertical CCN concentrations across entire above-cloud lidar curtains. These lidar-derived CCN concentrations can be used to evaluate model performance, which we illustrate using an example CCN concentration curtain from WRF-CAM5. These results demonstrate the utility of deriving vertically-resolved CCN concentrations from lidar observations to expand the spatiotemporal coverage of limited or unavailable in situ observations.

## 1 Introduction

One of the most pressing environmental questions is how Earth's climate will respond to anthropogenic emissions and associated radiative forcings. Natural and anthropogenic aerosols and their interactions with radiation and clouds play a key role in climate change and its uncertainty. Effective radiative forcing due to direct aerosol-radiation interactions (ERFari) include scattering and absorption of incoming solar radiation, while effective radiative forcing due to interactions between aerosols and clouds (ERFaci) is defined by the way that aerosols interact with clouds, and consequently, how clouds interact with radiation (Lohmann & Feichter, 2005; Andreae & Rosenfeld, 2008). These indirect effects include changes in cloud albedo and cloud lifetime, whose impacts on incoming solar energy can be significant in terms of temperature change at the surface and in the atmosphere (Budyko, 1969; Twomey, 1974; Albrecht, 1989; Andreae, 2009). Such aerosol-cloud interactions may have a large, but highly uncertain, cooling effect, as defined by the Intergovernmental Panel on Climate Change (IPCC) (Armour et al., 2021).

Uncertainty of aerosol-cloud interactions is especially high compared to other radiative forcings due in part to poor process-level understanding (Boucher et al., 2013). While the uncertainty remains high, estimates of ERFaci from observational and modelling studies have become more similar in the Sixth Assessment Report (AR6) than the Fifth (AR5), which resulted in a higher (negative) ERFaci magnitude (Forster et al., 2021). Factors that complicate observations of aerosol-cloud interactions include limited ability of non-polarimetric, passive satellite techniques to retrieve cloud and aerosol properties simultaneously in the same location, swelling of hygroscopic aerosols in high relative humidity (RH)/near-cloud environments, and effects of observational scale and meteorological context buffering responses of clouds to aerosol perturbations (Rosenfeld et al., 2014; Stevens & Feingold, 2009). High RH-induced swelling introduces artifacts into retrieval products, and cloud responses to aerosol perturbations are difficult to untangle using observations alone. Gaps in



fundamental understanding and sparse observations can also result in misrepresentation of aerosols in large-scale models (Boucher et al., 2013).

To better understand aerosol-cloud interactions, one key task is to improve the representation of cloud condensation nuclei (CCN) in forecasting models. CCN are the subset of aerosol particles that activate into droplets in ambient clouds. Their modulation can have a profound impact on cloud optical properties, microphysical evolution, and impacts on precipitation and climate (Andreae & Rosenfeld, 2008; Seinfeld et al., 2016). While in situ measurements of aerosols and CCN are critically important because they constrain aerosol properties at cloud top and base, they are most often limited to a

small spatiotemporal scale (Prather et al., 2008; Choudhury & Tesche, 2022). Since aerosols affect the planetary radiative balance on a global scale, we need other ways to obtain information about their concentrations and characteristics at greater spatial and temporal scales. One established approach to address this limitation is the usage of satellite and, to some extent, airborne in situ and remote sensing measurements of aerosol optical properties to constrain global aerosol distributions (Seinfeld et al., 2016). Although airborne measurements are generally limited to a small spatiotemporal domain, they can

better constrain aerosol and CCN distributions and, in combination with models and satellite observations, are a valuable constraint for aerosol distributions (Prather et al., 2008; Shinozuka et al., 2020).

Many studies have used remote sensing of aerosol optical properties to glean information about aerosol and CCN concentrations in different regions of interest (Ghan & Collins, 2004; Kapustin et al., 2006; Shinozuka et al., 2009; Shinozuka et al., 2015; Lv et al., 2018; Kacarab et al., 2020). While a significant amount of information can be obtained

from past remote sensing approaches, there are also limitations. For example, retrieving CCN concentration requires supplemental information, such as chemical composition (Petters & Kreidenweis, 2007), that is not always available or sufficiently accurate from remote sensing measurements (Kapustin et al., 2006; Shinozuka et al., 2009). One technical limitation is associated with hygroscopic uptake of water (and swelling) of aerosols, which increases satellite retrieved aerosol optical depth (AOD) but may not correspond to an increase in aerosol and/or CCN concentration, thus weakening the

relationship between both variables, as noted by Hasekamp et al. (2019). More reliable information about aerosol hygroscopicity and RH could improve CCN retrievals from satellite measurements (Kapustin et al., 2006; Jeong et al., 2007; Liu et al., 2007; Shinozuka et al., 2009).

Other CCN retrieval limitations lie in instrument capabilities and the relative size of CCN compared to the full spectrum of atmospheric aerosols. For example, while a large fraction of CCN should be captured by instruments with

channels in the visible and near-infrared part of the spectrum that can observe find mode aerosols, CCN at smaller ranges of the aerosol size distribution (i.e., 50-100 nm in diameter) (Meng et al., 2014) may be easier to capture using UV channels. Another common issue arises from heterogeneities of the aerosol vertical distribution profile, as passive remote sensing instruments (e.g., polarimeter, radiometer) provide column-effective products that cannot resolve vertical variations in aerosol or CCN properties, though polarimeters do have coarse sensitivity to aerosol location and can resolve aerosol size

distribution properties of fine and coarse mode aerosols. Active sensors, on the other hand, such as lidar, or combined lidar and polarimeter data sets have increased capability in measuring vertical profiles and have been used to derive aerosol



number concentrations (Schlosser et al., 2022) but are still subject to uncertainties and errors. For example, in Ghan et al. (2006), extinction and backscatter coefficients from Raman and micropulse lidar were used to retrieve CCN profiles, and the vertical heterogeneities in aerosol size distribution and composition were found to be the dominant source of error in
retrievals. A similar study by Lv et al. (2018) found that temporal heterogeneity of the atmosphere caused errors in CCN retrievals. Although some pioneering studies have applied Raman and micropulse lidar (Ghan & Collins, 2004; Ghan et al., 2006; Mamouri & Ansmann et al., 2016; Tsekeri et al., 2017; Marinou et al., 2019) and HSRL (Lv et al., 2018) to retrieve aerosol, CCN, and/or ice nucleating particle (INP) concentrations, significant assumptions have been made to mitigate inadequate information content from lidar alone for constraining aerosol properties, such as humidification factor remaining
constant with height and vertical distribution of extinction and backscatter coefficients being identical to the vertical distribution of CCN.

We use observations made during the NASA ObseRvations of Aerosols above Clouds and their intEractionS (ORACLES) campaign that took place between 2016 and 2018 over the Southeast Atlantic (SEA) (Redemann et al., 2021). This region is of particular interest and importance due to a seasonal cycle from July to October of biomass burning
emissions that are advected westward atop a semi-permanent deck of marine stratocumulus clouds. Most of these smoke aerosols are lofted above and separated from a large stratocumulus deck. At different times and locations they can be entrained into the boundary layer, directly interacting with clouds (Kaufman et al., 2003; Ross et al., 2003; Adebiyi et al., 2015; Zuidema et al., 2016). Moreover, stratocumulus clouds have a significant impact on global climate and are poorly represented in climate models as being too few in quantity and too bright (Bony & Dufresne, 2005; Nam et al., 2012),
although some recently developed models do now appear to represent both the distribution of these clouds and their response to temperature change more realistically (Cesana et al., 2019; Tselioudis et al., 2021). Non-polarimetric passive remote sensing of aerosols also becomes more difficult in the presence of low stratocumulus clouds (Coddington et al., 2010; Chang et al., 2021). For these reasons, our ability to accurately predict CCN concentrations and represent them in models becomes more important for the SEA. One major objective of ORACLES was to obtain the observational constraints and testbed for
future climate model and biomass burning aerosol (BBA) remote sensing algorithm development (e.g. Mallet et al., 2019; Xu et al., 2021; Doherty et al., 2022; among others). The goal of this study is to use the unique and novel ORACLES data set to develop relationships between HSRL-2 observables and in situ CCN concentrations within the smoke plume to obtain vertically-resolved CCN concentrations throughout a region dominated by BBA.

Developing a method to obtain accurate CCN concentrations from lidar observables could greatly aid in evaluating
CCN concentration in global and regional climate models, which is a key variable for determining aerosol-cloud interaction mediated radiative forcing. In addition, this study is relevant to the National Aeronautics and Space Administration (NASA) Atmosphere Observing System (AOS) mission regarding the improvement of retrievals of CCN concentration to reduce indirect forcing uncertainties in climate models. With plans for a future spaceborne HSRL in AOS, it is highly beneficial to develop methods to enable future use of a satellite-based HSRL to infer vertically-resolved CCN concentrations. The
methodology described here, while specific to BBA in the SEA, will lay the groundwork for future analyses to determine



relationships to derive CCN for additional aerosol types. The paper is organized as follows: In Section 2 we briefly discuss data collocation and filtering techniques. In Section 3 we investigate the relationships between CCN concentration and HSRL observables. Comparison of the results to a previous study and discussion about the applicability of the method is given in Section 4, followed by a further comparison of resultant estimates of CCN concentrations to WRF-CAM5 model

output.

## 2 Data and Methods

ORACLES focused on filling an observational gap regarding aerosol and cloud properties to improve climate model representation of aerosol-cloud interactions. These observations were made using a combination of remote sensing and in

situ instruments located on the NASA P-3 (2016-2018) and ER-2 (2016 only) aircraft. As indicated by the flight tracks in Figure 1, deployments were based in Walvis Bay, Namibia in September 2016 and São Tomé and Príncipe in August 2017 and September-October 2018. The methodology proposed here, as well as the resultant parameterized equations can be further used to produce CCN profiles from lidar observations.

The three primary observations of interest in this study include in situ measured CCN concentration, and HSRL-2

backscatter and extinction. In total there are 10 campaign days where all data sets overlap under our collocation criteria constraints described in Section 2.2. Instrument details are given in Section 2.1 and summarized in Table 1.

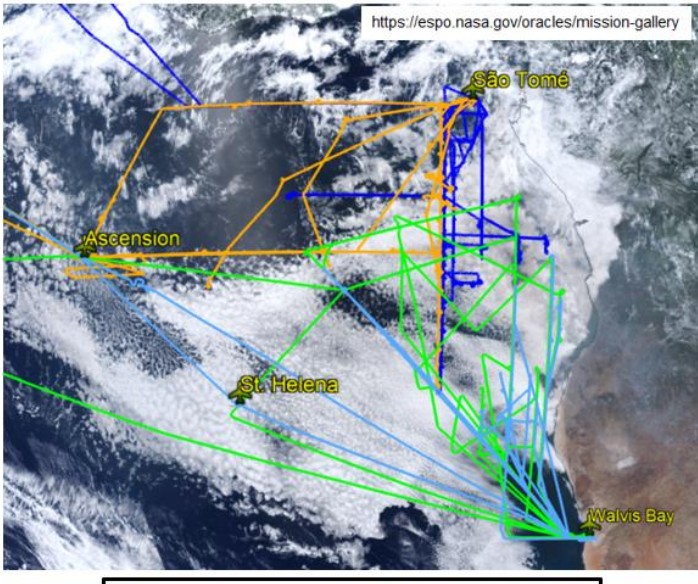

**Figure 1: ORACLES flight tracks over the Southeast Atlantic color-coded by year and aircraft.**



### 2.1 Instrumentation

#### 2.1.1 HSRL-2

The NASA Langley Research Center HSRL-2 measures aerosol backscatter and depolarization at 355, 532, and 1064 nm and aerosol extinction via the HSRL technique at 355 and 532 nm (Shipley et al., 1983; Burton et al., 2018). Aerosol extinction is also provided at 1064 nm from the product of aerosol backscatter at 1064 nm and an inferred lidar ratio at 1064 nm. The HSRL-2 measurement technique uses the spectral distribution of the return signal to distinguish between aerosol and molecular returns. This means that aerosol backscatter and extinction coefficients are determined independently as opposed to being determined based on a lidar ratio assumption typically used in elastic backscatter lidar retrievals (Hair et al., 2008). We utilize HSRL-2's products of particulate backscatter and extinction at 355 and 532 nm. Compared to HSRL-1, HSRL-2's additional measurement channel at 355 nm may theoretically be expected to have higher sensitivity to smaller particles, such as CCN, that are especially relevant in aerosol-cloud interactions (Burton et al., 2018). The horizontal and vertical resolutions of aerosol backscatter and depolarization are approximately 2 km and 15 m, respectively. The horizontal and vertical resolutions of aerosol extinction coefficients are approximately 12 km and 300 m, respectively, but extinction profiles are interpolated to match the finer resolutions matching those of backscatter and depolarization. The temporal resolution of aerosol backscatter and extinction coefficients are approximately 10 s and 60 s, respectively. However, the exact temporal depends on aircraft speed. Uncertainty in the lidar observables depends on contrast ratio and aerosol loading, among other factors, but uncertainties within 5% can be achieved under certain conditions (Burton et al., 2018).

Another way of using the HSRL-2 extinction coefficient is through calculation of aerosol index (AI). AI is the product of the Angstrom exponent ($\alpha$) and AOD and is a column-effective parameter commonly used as a proxy for CCN concentration (Liu & Li, 2014; Rosenfeld et al., 2014; Stier, 2016). AI is typically thought to represent concentrations of small particles better than other optical properties due to the Angstrom exponent containing information on particle size (Bréon, 2002; Liu et al., 2007). We calculate an AI by first calculating the Angstrom exponent via Eq. (1):

$$\alpha = - \frac{ln[EXT(\lambda_1)/ EXT(\lambda_2)]}{ln (\lambda_1/\lambda_2)} \tag{1}$$

and then multiplying it by HSRL-2 extinction (EXT) at both 355 and 532 nm via Eq. (2):

$$AI = \alpha * EXT \tag{2}$$

resulting in one AI value for each of these two channels with HSRL capability. Each AI value is then multiplied by the vertical collocation bin depth for the respective year (Table 2). This calculation is performed for each individual HSRL-2 profile and values used for analysis are averages that result from the data collocation process.



### 2.1.2 Georgia Institute of Technology (GIT) CCN Instrument

The other primary instrument and data set used in this study is the Georgia Institute of Technology (GIT) Droplet
Measurement Technologies (DMT) CCN counter (CCN-100). It measures in situ CCN concentration at various levels of
water vapor supersaturation (S), here between 0.1% and 0.4% (Kacarab et al., 2020, Redemann et al. 2021). The instrument
is designed as a continuous-flow streamwise thermal-gradient chamber (CFSTGC; Roberts & Nenes, 2005). In this type of
system, quasi-uniform supersaturation is generated at the centerline of a cylindrical flow chamber, owing to the continuous
transport of heat and water vapor from wetted walls subject to a temperature gradient. The difference in heat and water vapor
diffusivity in the radial direction ensures that supersaturation is generated – with levels that depend on the flow rate and
temperature gradient. The continuous flow feature allows for quick sampling, with roughly 1 Hz frequency (Roberts &
Nenes, 2005), which is critical for rapidly changing environments such as those encountered in airborne sampling. Aerosols
that activate into droplets with radius greater than 0.5 μm are counted as CCN at the end of the growth chamber. During
ORACLES the horizontal resolution of in situ observations depends on aircraft speed. Uncertainty associated with CCN
number concentration is ±10% at high signal-to-noise ratio (S/N) and 5-10 cm$^{-3}$ at low S/N. Supersaturation uncertainty is ±
0.04% (Rose et al., 2008).

### 2.1.3 HiGEAR Instrument Suite

Dry extinction coefficients and observations of mass-to-charge (m/z) ratio of 44 relative to total organic aerosol
concentration (f44) data sets come from the Hawaii Group for Experimental Aerosol Research (HiGEAR) instrument suite.
In situ dry extinction coefficients are calculated using data measured by two TSI 3653 nephelometers and two Radiance
Research particle soot absorption photometers (PSAPs) (Shinozuka et al., 2020; Redemann et al., 2021). Aerosol light
scattering coefficients are measured by nephelometers at 450, 550, and 700 nm and then interpolated to and reported at the
PSAP light absorption wavelengths of 470, 530, and 660 nm. Dry extinction is calculated using the sum of scattering and
absorption coefficients. The resulting extinction coefficients are then linearly interpolated to 500 nm to compare ORACLES
campaign average results against results from Shinozuka et al. (2015) in Section 4.

        The f44 data is measured using the Aerodyne high-resolution time-of-flight (HR-ToF) aerosol mass spectrometer
(AMS) and can be used to estimate aerosol age (Cubison et al., 2011; Dobracki et al., 2022). This instrument provides
quantitative size and chemical mass loading information for non-refractory sub-micron aerosol particles. The f44 data set is
available for all days of overlap between CCN and HSRL-2 observations except for 20170812 and 20170828, where there
are no salvageable AMS data.

**Table 2: List of instruments and data sets used in this study, as well as their respective resolution, measurement type, and aircraft**
**location.**



| Instrument | Variables | Resolution (Temporal / Vertical) | Measurement Type | Aircraft |
|---|---|---|---|---|
| High Spectral Resolution Lidar 2 (HSRL-2) | Aerosol backscatter coefficient (355 and 532 nm), extinction coefficient (355 and 532 nm) | 2016-2017: 10 s / 15 m for backscatter; 60 s / 315 m for extinction (reported at 15 m) 2018: 10 s / 15 m for backscatter; 60 s / 150 m for extinction (reported at 15 m) | Remote Sensing | ER-2 (2016) P-3 (2017-2018) |
| Cloud condensation nuclei (CCN) counter (DMT CCN-100) | CCN number concentration at different supersaturations (S) | 1 s | In Situ | P-3 |
| HiGEAR Particle Soot Absorption Photometer (PSAP) and nephelometer | Absorption (470, 530, 660 nm) and scattering coefficients (450, 550, 700 nm) | 1 s | In Situ | P-3 |
| Edgetech 3-Stage Hygrometer | Ambient relative humidity (RH) | 1 s | In Situ | P-3 |
| HiGEAR Aerodyne HR-ToF aerosol mass spectrometer (AMS) | Mass-to-charge ratio m/z 44 relative to total organics (f44) | 1 s | In Situ | P-3 |



## 2.2 Data Collocation and Filtering

During the ORACLES campaign, HSRL-2 profiles were observed at high-altitude, above-plume flights legs, and CCN
concentration was measured in situ at lower altitudes in and around the smoke plume. Each data set has a different spatial
and temporal resolution, so collocation in time and space is necessary before correlation analysis can be performed. One
important consideration here is the difference in aircraft set-up between 2016 and 2017-2018. In September 2016 the ER-2
flew high altitude legs and carried the HSRL-2 while in situ instruments were located on the P-3, which flew at lower
altitudes. However, in August 2017 and September-October 2018 both HSRL-2 and the in situ instruments were deployed on
the P-3. In these the deployments, the P-3 often flew at above-plume altitudes to optimize sampling by the HSRL-2 and other
remote sensing instruments and later sampled at lower altitudes for in situ observations. Therefore, there is a slightly longer
time gap between observations from these instruments in the 2017 and 2018 deployment years.

        The result of our data collocation technique is a one-to-one comparison between averaged CCN concentration
values and averaged HSRL-2 observations, both observed in approximately the same time and space defined using three
independent collocation criteria, as follows. For any given HSRL-2 profile, the collocation method finds CCN concentration
measurements that fall within a set amount of time (dt) from when the HSRL-2 profile was measured, within a set horizontal
distance (dd) from the profile, and within set vertical bins (dh). Observations that remain after each of these criteria have
been applied are then averaged to allow for a one-to-one comparison. A schematic of this process is shown in Figure 2. Note
that while each year uses the same collocation method, 2017 and 2018 have a different aircraft set-up that results in a longer
time gap between measurements (denoted by t + Δt).

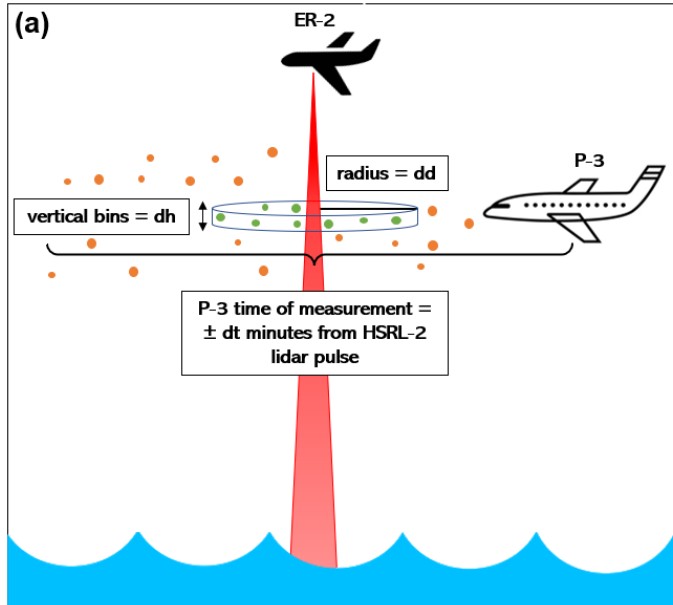
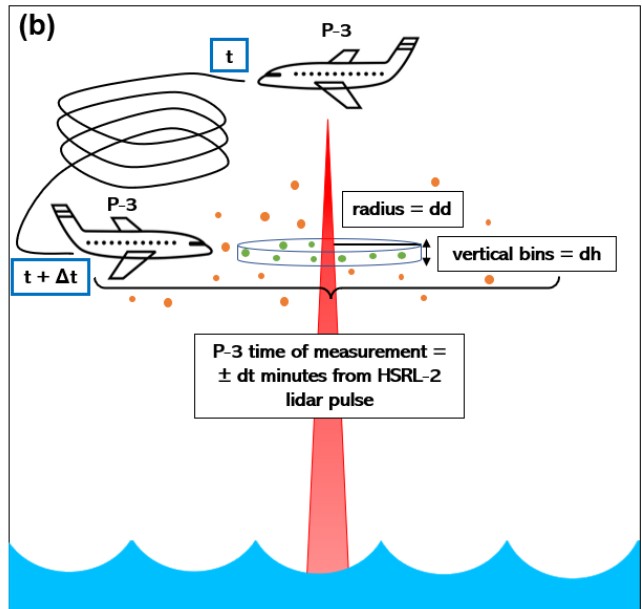



**Figure 2: Graphic depicting data collocation process for a) 2016 and b) 2017-2018. CCN concentration measurements that fall within the time, horizontal distance, and vertical bin criteria (green points) are averaged to compare to the average HSRL-2 backscatter and extinction coefficients that fall within the same vertical bin.**


      Different collocation criteria values were sensitivity tested to minimize the effects of spatiotemporal variability on the correlation between HSRL-2 observables and in situ CCN concentrations while also selecting a representative portion of the data set to analyze. These sensitivity tests were performed by varying the horizontal, vertical, and temporal criteria one at a time and evaluating their impacts on correlations between CCN – HSRL-2 backscatter and extinction. The final chosen

values are given in Table 2 and will be used for all subsequent results. We hold horizontal distance constant at ±1.1 km for all years to approximately correspond to the horizontal distance over which the HSRL-2 products are aggregated, and to avoid averaging multiple HSRL-2 profiles in the horizontal. As a result, each average in situ CCN concentration value corresponds to a vertical average of HSRL backscatter and extinction coefficients from a single profile. Vertical bin sizes vary between years due to slight adjustments made to optimize the correlation, and temporal collocation allowances are

increased for 2017 and 2018 due to the instruments' deployment on the same aircraft resulting in a longer time gap between high-altitude remote sensing and low-altitude in situ observations.

      Another pre-analysis step is filtering the collocated data set by RH and supersaturation (S) thresholds. While one goal of the data filtering step is consistency between each year of the analysis, a few differences in instrument settings and data availability make it necessary to slightly alter the RH and S thresholds between each year. We performed sensitivity

tests for different values and ranges of RH and S, and final values are given in Table 2. In general, the sensitivity tests suggest that observations made at high ambient RH tend to weaken the CCN versus backscatter and extinction relationships due to swelling of highly hygroscopic aerosols that causes an increase in backscatter and extinction without a corresponding increase in CCN concentration. Testing of different thresholds suggests that we can mostly avoid this hygroscopic effect by filtering out observations made at RH > 40%. Due to a limited amount of collocated data points in 2018, caused primarily by

remote sensing and in situ observations from a single aircraft, the RH threshold is increased to 50% for 2018 only. A threshold of 40% resulted in a very small resultant RH range (34-39%) that corresponded only to low CCN concentrations. Therefore, the threshold was increased to 50% to allow for more data points and a relationship over a range of CCN concentrations more like that for 2016 and 2017. Importantly, since our focus is on CCN within the smoke plume, this constraint still retains most of the data set (Table 2). However, for future analyses focused on boundary layer (relatively high

ambient RH) CCN, it will be important to correct lidar measurements made at higher RH to avoid eliminating those CCN concentrations most relevant for aerosol-cloud interaction studies. In terms of S, Köhler theory (Köhler, 1936) suggests that higher S values allow for more aerosols to activate into cloud droplets. Therefore, at lower S values there may be high aerosol concentrations and lidar backscatter and extinction coefficients but decreased in situ CCN concentrations. Since slightly different values and ranges of S were used by the CCN counter each year, the resultant filtering thresholds also vary

accordingly.





Another goal of data collocation and filtering steps is to maintain as representative a collocated data set as possible. As shown in Table 2, our RH and S thresholds represent between 52-76% of the total collocated data set. Amidst data limitations, accounting for two different aircraft set-ups, and working to maximize correlations, these percentages are reasonably representative of the total data set. Another important note in relation to both the location of our collocated data

points within the smoke plume and the RH filtering is the inherent exclusion of marine boundary layer (MBL) CCN concentrations. This is in part due to the nature of the SEA environment and the limitation of having few lidar observations below the semi-permanent stratocumulus deck. We thus focus on developing a relationship specifically for HSRL-2 observables and BBA above-cloud within this region. Though below-cloud CCN concentrations are of most interest in terms of aerosol-cloud interaction studies, links have also been found between above-cloud aerosol and cloud microphysical

properties (Gupta et al., 2021), validating the need for increased information about above-cloud CCN concentrations as well. Furthermore, CCN concentrations were found to be significantly higher above-cloud in the free troposphere than those in the MBL (Redemann et al., 2021), again validating a focus on analyzing above-cloud CCN concentrations using HSRL-2 observables.

**Table 2: Collocation criteria, relative humidity, and supersaturation thresholds chosen to analyze each year of ORACLES. Percentage of the entire dataset represented by each relative humidity threshold, percentage of collocated data set represented by each supersaturation threshold, as well as final dates and quantity of data points are also listed.**

|  | **2016** | **2017** | **2018** |
|---|---|---|---|
| horizontal criteria (dd) | ± 0.01° | ± 0.01° | ± 0.01° |
| vertical bin size (dh) | 45 m | 60 m | 75 m |
| time criteria (dt) | ± 0.1 hr | ± 0.3 hr | ± 0.2 hr |
| relative humidity (RH) | RH ≤ 40% | RH ≤ 40% | RH ≤ 50% |
| supersaturation (S) | S = 0.30% | 0.22% ≤ S ≤ 0.34% | 0.23% ≤ S ≤ 0.40% |
| % of data measured below RH threshold (with CCN available) | 75% | 60% | 52% |
| % of collocated data set in S range | 76% | 72% | 66% |
| # of days represented in collocated data set (after RH and S filtering) | 1 (20160912) | 3 (20170812, 20170815, 20170828) | 6 (20180927, 20181002, 20181007, 20181010, 20181019, 20181023) |
| # of data points (after RH and S filtering) | 40 | 13 | 27 |



# 3 Results

## 3.1 CCN versus HSRL-2 Extinction and Backscatter

Following the data collocation and filtering, we analyze the correlation between CCN concentration and HSRL-2 observables. Year-by-year analyses were done to determine ideal collocation criteria, as described in Section 2.2. However, the primary goal of this study is to develop a relationship between CCN and HSRL-2 backscatter and extinction coefficients. Therefore, we need to show that year-by-year analyses can be combined in a way that maintains the overall strength of the linear relationships. Figure 3 combines all collocated data points from September 2016, August 2017, and September-October 2018 to fit a relationship between CCN concentration and HSRL-2 backscatter and extinction at 355 and 532 nm. This complete data set is represented by 80 total data points spanning 10 days.

We show the Pearson correlation coefficient (R), Spearman rank correlation coefficient (in parentheses), root mean square error (RMSE), and percentage of data within ±20% of the linear regression line. The combination of data from different measurement periods across three years results in strong correlations (0.95-0.97) between both variables. This result suggests that our collocation and filtering methodology is reasonable and holds well for multiple observational periods. In addition, different symbols designating different years support the representativeness of the collocated data set, as no one period of observations completely stands out from another. RMSE is on the order of 100 cm$^{-3}$, with values slightly lower for backscatter than extinction coefficients. The amount of data within ±20% of each respective linear regression line ranges from 74-82%. These values, together with RMSE, suggest a relatively low amount of scatter around the regression line for both wavelengths and coefficients. Using Eq. (3):

$$EXT = Q_e N\pi r^2 \tag{3}$$

where $Q_e$ is extinction efficiency (approximated as 2), N is number concentration, r is radius, and EXT/N is found using the regression slopes in Figure 3, an approximate CCN radius is calculated to be about 0.15 μm, with an effective cross-section of around 0.1 μm. These values are on the order of what is physically expected for smoke aerosols.



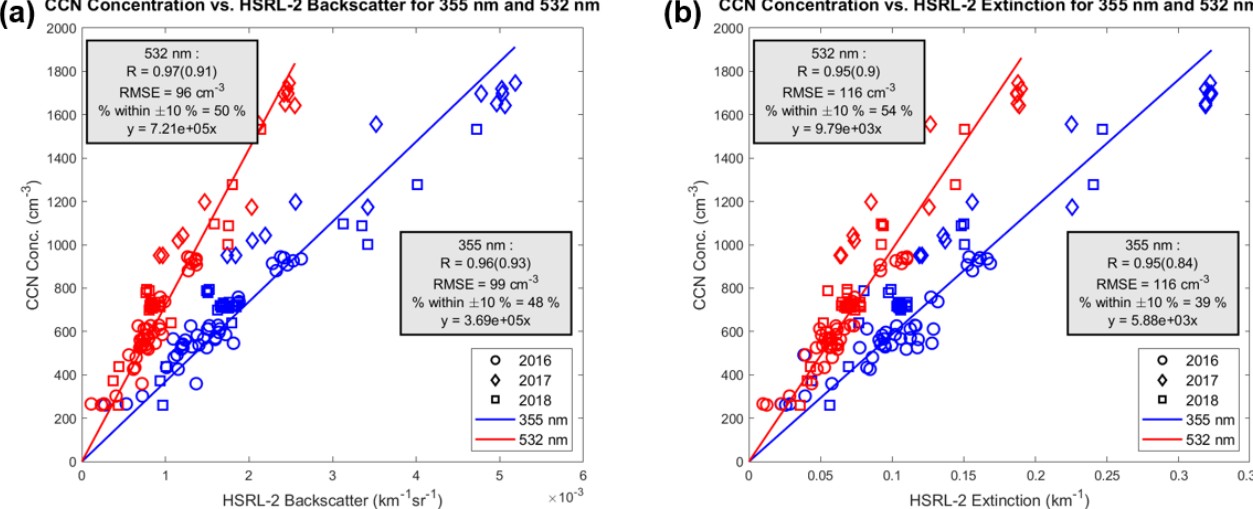

**Figure 3: CCN concentration vs. HSRL-2 backscatter and extinction coefficients, with both wavelengths shown on each panel. This combined data set represents 10 days of observations and 80 total collocated data points (per coefficient), covering all three years of ORACLES. Supersaturation for these observations ranges between 0.22-0.4%. Lines of best fit are forced through the origin to represent practicality of using linear regression equations to quantitatively obtain CCN concentrations using HSRL-2 observables. The Pearson correlation coefficient is shown, with Spearman rank correlation coefficient given in parentheses.**

## 3.2 CCN versus Aerosol Index

Another parameter that we explore in relation to CCN concentration is AI, calculated using HSRL-2 extinction coefficients as described in Section 2. Results found using AI for the three-year combined data set are shown in Figure 4. Overall, the general trends found in these results match those seen in Figure 3 for the CCN concentration versus HSRL-2 correlation. Again, these relationships hold for the three-year combined data set without any one year appearing as an outlier. We again show the Pearson correlation coefficient (R), Spearman rank correlation coefficient (in parentheses), root mean square error (RMSE), and amount of data within ±20% of the linear regression line. AI calculated using extinction at 532 nm relates slightly more strongly with CCN concentration than does AI calculated using the 355 nm coefficient. This difference could be due to slightly higher uncertainty of HSRL-2 extinction at 355 nm than 532 nm (Burton et al., 2018). RMSE is slightly higher than in Figure 3, and the amount of data within ±20% of the linear regression line ranges from 39-49%, indicating slightly more scatter than when using HSRL-2 backscatter and extinction coefficients alone. This could, in part, be due to the very small range of AI values present in the comparison. Nevertheless, the relationships given in Figures 3 and 4 are comparable and serve as good indicators for representing CCN concentration.

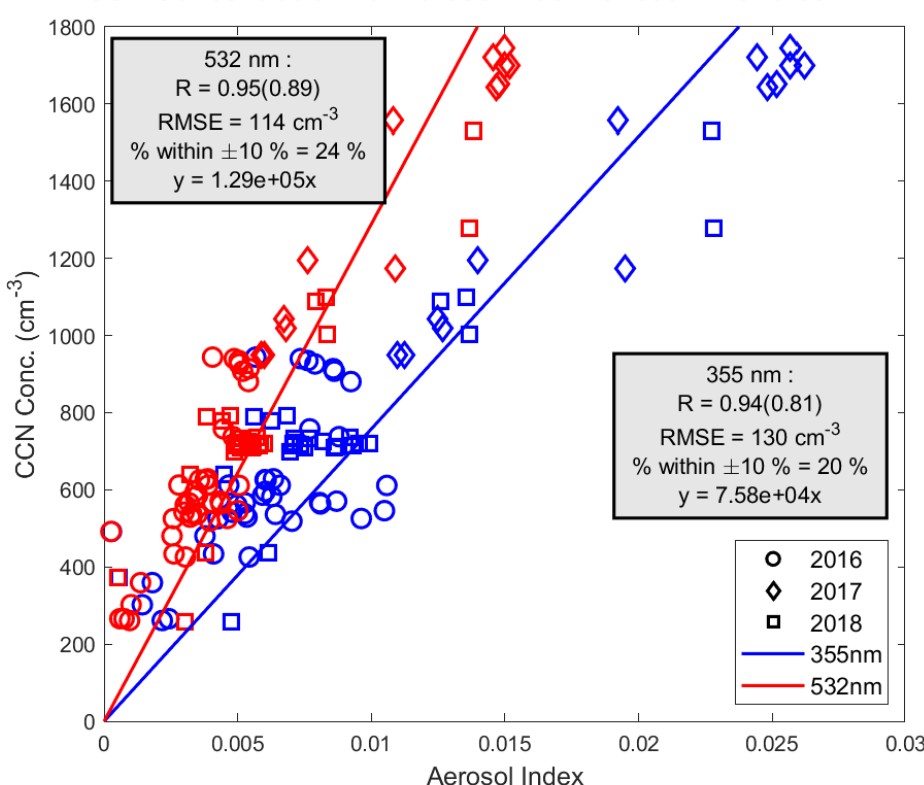

**Figure 4: CCN concentration vs. aerosol index is given for the combined three year data set. Blue scatter points show results calculated using extinction at 355 nm and red scatter points show results calculated using extinction at 532 nm. Supersaturation for these observations ranges between 0.22-0.4%. Lines of best fit are forced through the origin to represent practicality of using linear regression equations to quantitatively obtain CCN concentrations using lidar observables. The Pearson correlation coefficient is shown, with Spearman rank correlation coefficient given in**
**parentheses.**

The singular difference between the extinction coefficient and AI is multiplication by the Angstrom exponent, an indicator of particle size. There is almost no difference found between how well extinction and AI relate to CCN concentration. The fundamental similarities in both relationships suggest that there is very little variation in the range of

345 Angstrom exponents that multiply the aerosol extinction coefficient. Therefore, there is likely only small variation in the size of aerosol being measured. Since observations included in this analysis focus on those made in the smoke plume, this result is reasonable. We are not performing this analysis for a variety of aerosol types and different sizes, but instead focus on the small range of BBA properties within the smoke plume.



## 4 Discussion

Section 3 focused on the linear relationships between CCN concentrations and HSRL-2 backscatter and extinction coefficients over the BBA-dominated SEA. We also investigated the possibility of using AI as an additional parameter of comparison. In Section 4, we will use in situ "dry" extinction coefficients to explore the overlap between our campaign-averaged values with those from other regions and aerosol types (Shinozuka et al., 2015) and will analyze some large-scale implications and applications of this work.

### 4.1 Observed Relationships

Overall, data collocation and filtering methods suggest that for time intervals between ±6-18 minutes, horizontal distance separations within about 2 km, and low ambient RH, CCN concentration relates very strongly with lidar observables, with Pearson correlation coefficients of 0.95 to 0.97. The relationship between in situ CCN concentrations and HSRL-2 backscatter coefficients is slightly stronger than that using extinction coefficients, potentially due to the higher uncertainties and coarser resolutions associated with the extinction coefficient (Burton et al., 2016). Nevertheless, we find that both backscatter and extinction are positively and linearly correlated with in situ CCN concentrations. Resultant regression equations allow us to use lidar observables alone to estimate CCN concentrations for this aerosol type, as further illustrated in Section 4.3.

We also considered AI, as calculated from HSRL-2 extinction coefficients, with the expectation that an aerosol property that implicitly contains information on aerosol size may correlate better with CCN concentrations than an optical coefficient alone. However, corresponding relationships with of in situ CCN concentration to AI were similar to those found using HSRL-2 extinction coefficients alone. Hence, we do not find in this study that AI is a better parameter to represent CCN concentration than other optical properties as other studies have suggested (Liu & Li, 2014; Rosenfeld et al., 2014; Stier, 2016). Rather, we find relationships between CCN and AI are nearly identical to those of the HSRL-2 observables. This finding suggests minimal variation in the Angstrom exponent and implies that our analysis focuses on a small size range of aerosol observed within the smoke plume.

We hypothesized that HSRL-2 observables at 355 nm would be more strongly related to observed CCN concentrations due to a smaller wavelength interacting with smaller aerosols. However, this was largely not the case. Many of our year-by-year analyses resulted in a slightly stronger relationship between CCN concentration and HSRL-2 backscatter and extinction at 532 nm. For looking at the three-year combined data set, both wavelengths result in an identical correlation coefficient (Figure 3). This discrepancy will be explored in future analyses using theoretical calculations of optical properties from Mie theory and CCN concentrations derived from κ-Köhler theory (Petters & Kreidenweis, 2007). However, results from Burton et al. (2016) in relation to the information content and sensitivity of a 3β + 2α lidar such as HSRL-2 suggest a very minor difference in the sensitivity of each wavelength to aerosols with small radii, such as CCN. Therefore, our



preliminary literature search into this discrepancy suggests that it may be reasonable to find little variability in the strength of the relationships between lidar observables at both wavelengths and observed CCN concentrations.

### 4.2 Representativeness of Collocated Data Set

In Figure 3, relationships between in situ CCN concentration and HSRL-2 backscatter and extinction coefficients were shown for the three-year combined data set. While the strength of these relationships for the multi-year data set suggests that we can use this method to analyze BBA found in the ORACLES data, it is important also to recognize inherent environmental differences for each of these years. First, for each of the three years of this campaign, observations were made during different parts of the seasonal biomass burning cycle that occurs between June and October (Redemann et al., 2021). This difference primarily impacts the age of aerosols observed during each year, therefore potentially affecting CCN activity. Secondly, observations were made in different regions of the SEA each year. While we have shown that these differences do not appear to significantly hinder the construction of a three-year combined data set for BBA, we will examine these differences in aerosol age and location of measurement to understand the extent of data considered when all three years are analyzed together.

One commonly occurring trajectory for BBA in this region suggests that the plume core is lofted over the SEA near the equator, whereby the aerosols move westward, following a spiral path, and later descend into the boundary layer near 20°S and close to the coast (Redemann et al., 2021). Though the exact trajectory changes based on month within the seasonal burning cycle, the overall pattern implies that aerosols measured farther north within the SEA domain are likely younger, while aerosols observed farther west and south tend to have travelled longer along the spiral pathway and are older aerosols. Comparing this conceptual model with the approximate locations of our collocated data points (Figure 5) suggests that observations from 2017 and 2018 that lie farther north and west within the domain were likely measured closer to the time they were ejected from the smoke plume. In contrast, observations from 2016 lie farther south and closer to the coast, suggesting that they may be older. This conceptual model can be evaluated using measurements of f44. Higher values indicate increased amounts of carboxylic acids and imply that measured aerosols are relatively old. Using three-year combined CCN concentration versus extinction at 355 nm as an example, we color-code data by values of f44 (Figure 6). In general, 2017 and 2018 have lower f44 values ranging from 0.17-0.21, indicating slightly younger aerosols. Data from 2016 have a wider range of f44 values between 0.2-0.27, indicating some ages close to those from 2017 and 2018, and some that are older. The f44 data suggest that determinations of aerosol age may not always be as straightforward as depicted by the expected trajectory pathway (Figure 5).

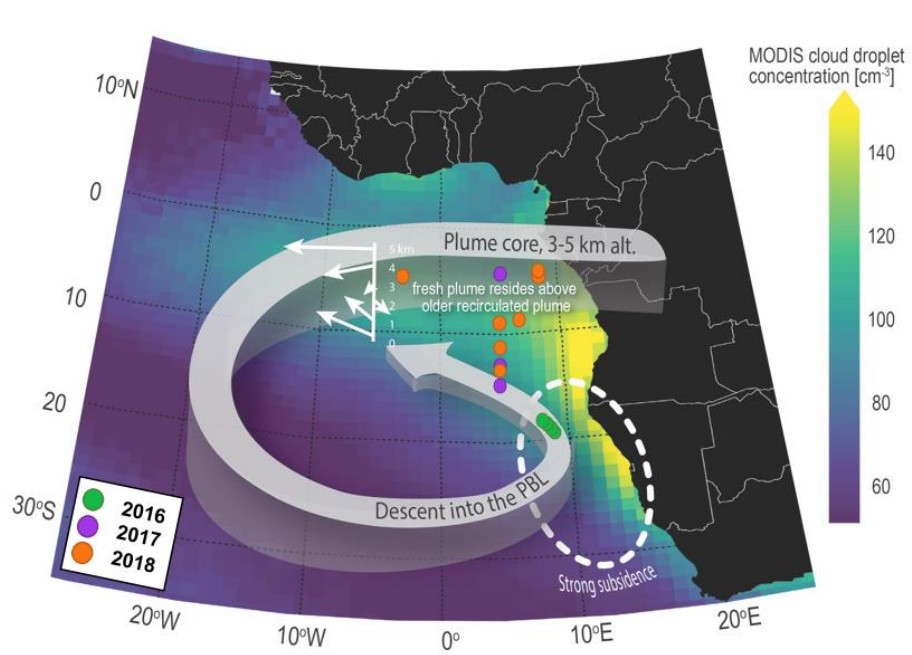

**Figure 5: Adapted from Redemann et al., (2021), this schematic shows the commonly occurring trajectory of smoke aerosols ejected over the Southeast Atlantic. Overlaid are approximate locations of collocated observations from 2016 (green), 2017 (purple), and 2018 (orange). By comparing their locations to the schematic, we hypothesize that the collocated observations made in 2016 are older aerosols, while collocated observations of aerosols in 2017 and 2018 are likely fresher and more recently ejected from the smoke plume.**





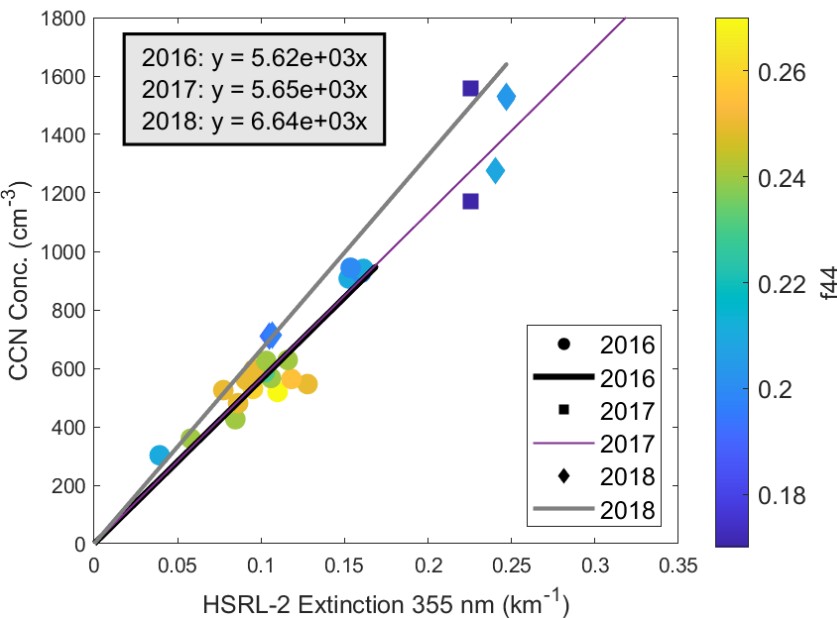

**Figure 6: CCN concentration vs. HSRL-2 extinction coefficient at 355 nm for 2016 (circles), 2017 (squares), and 2018 (diamonds) collocated data sets. Supersaturation for these observations ranges between 0.22-0.4%. The scatter points are color-coded by f44 value. The f44 values range from 0.2-0.27 for 2016, 0.17 for 2017, and 0.19-0.21 for 2018. The number of data points is lower than the full number of full collocated data points due to missing f44 data. Lines of best fit are forced through (0,0) to represent the practicality of using linear relationships to quantitatively obtain CCN concentrations from lidar observables.**

Lastly, we look at observations from ORACLES in the context of other regions and aerosol types to explore the representativeness of our collocated data set. Specifically, we compare our results to those from Shinozuka et al. (2015), a study focusing on similar relationships between CCN concentration and in situ aerosol extinction coefficient of dried particles. To compare our results most accurately with those from Shinozuka et al. (2015), we use a campaign-average value for CCN concentration and in situ "dry" extinction coefficient at 500 nm, as calculated from in situ measured particle scattering and absorption coefficients. These observations were made on the same instrument rack as CCN concentration and represent the extinction coefficient for dried particles pumped into aircraft instrumentation and unimpacted by ambient RH, unlike observations made by HSRL-2. This comparison is given in Figure 7, which is modified from Figure 6 in Shinozuka et al. (2015), with results from the current study added and labelled as "Southeast Atlantic." We overlay the campaign-average CCN concentration and 500 nm dry extinction coefficient as a scatter point, without altering the original regression lines. Campaign-average values were calculated using the entirety of the ORACLES in situ dry extinction and CCN concentration data sets (not limited by our collocation technique). The data were only filtered to exclude low CCN concentrations (CCN < 100 cm$^{-3}$) collocated with low extinction coefficients (in situ extinction < 0.02 km$^{-1}$) characteristic of a gap region and to match the S range (0.3-0.5%) from Shinozuka et al. (2015). This filtering was performed to provide as





close of a comparison as possible. Horizontal and vertical bars for each point represent the standard deviation. Overall, we find that our campaign-average CCN and in situ dry extinction values, specifically for the BBA-dominated SEA region, compare well with the relationships given for other regions and aerosol types. This comparison serves as another source of validation for the methodology of this study and suggests that performing a similar CCN concentration versus HSRL-2 observable relationship analysis for other regions and campaigns with different dominant aerosol types would be feasible.

Therefore, while the observations used in this study do not span the entire SEA, we have reason to believe that they account for different parts and ages of the smoke plume as it is advected westward. Since the in situ and lidar observations can be combined in a three-year data set that maintains a strong linear relationship between CCN concentrations and HSRL-2 observables, we believe that these results are representative of BBA in the SEA.

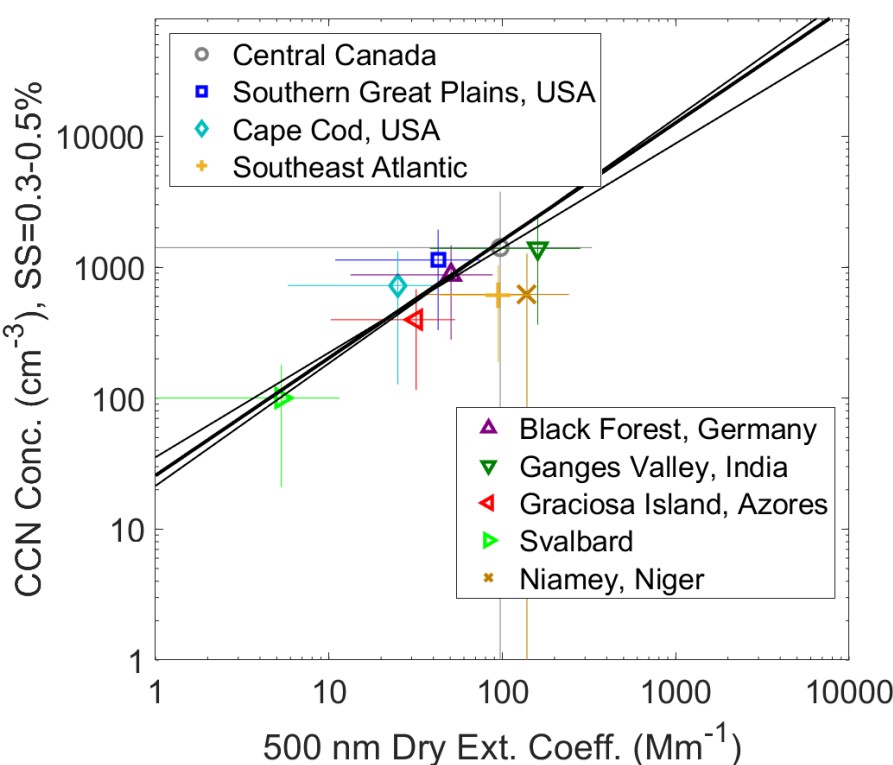

**Figure 7: Adapted from Figure 6 in Shinozuka et al. (2015), campaign-average in situ CCN concentration and in situ dry extinction at 500 nm are overlaid, using the label "Southeast Atlantic." The error bars are calculated using standard deviation, and the original best fit lines have not been altered. The supersaturation (SS) for these observations ranges from 0.3-0.5% to match the analysis done in Shinozuka et al. (2015).**





## 4.3 Application and Future Work

A goal of this work is the development of a method to derive CCN concentrations using HSRL-2 observations alone, and particularly to increase knowledge of the vertical distribution and variation of CCN. Below, we demonstrate this ability for one ORACLES flight leg, where we derive CCN concentrations for an entire HSRL-2 curtain using the regression equation derived from the three-year combined data set of in situ CCN concentration versus backscatter at 532 nm developed using a supersaturation range of 0.22-0.4% (Figure 3). Figure 8a shows calculated CCN concentrations from an above-cloud August 15, 2017 flight leg. Figure 8b shows CCN concentrations for this same flight leg simulated by the Weather Research and Forecasting model coupled with physics packages from the Community Atmosphere Model version 5 (WRF-CAM5; Shinozuka et al., 2020) at 0.5% supersaturation. Some features of our derived CCN concentrations agree reasonably well with the corresponding WRF-CAM5 output. However, our lidar-derived method results in a better estimate of smoke plume depth and altitude, cloud top heights, and small-scale variation of CCN concentrations, possibly due to turbulence and entrainment effects that are not yet captured by the model. Other model-observation comparison studies have shown that models such as WRF-CAM5 (and others) also tend to underestimate the altitude of the smoke plume (Shinozuka et al., 2020; Doherty et al., 2022).

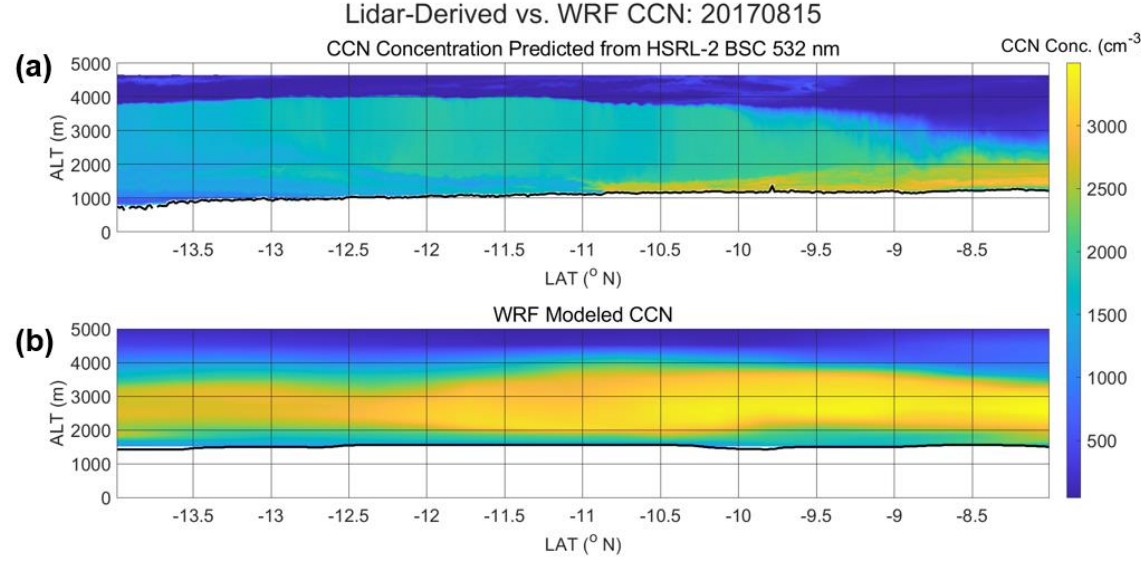

**Figure 8: (a) Derived curtain of CCN concentrations from the HSRL-2 backscatter coefficient at 532 nm for an ORACLES P-3 track on 15 August 2017. The black line corresponds to HSRL-2 cloud top height altitudes. (b) WRF-CAM5 model CCN curtain corresponding to the same P-3 track. The black line corresponds to the nearest HSRL-2 - interpolated altitude at WRF-estimated cloud top height.**



One potential reason for differences in the magnitude of CCN concentrations between WRF-CAM5 and our lidar-derived method includes the tendency of WRF-CAM5 and other models to represent the plume as more vertically diffuse than is generally observed (Doherty et al., 2022). This overly diffuse representation of the smoke plume can also result in an overestimation of aerosols lying above the stratocumulus deck and that get mixed into clouds, further promoting an

overestimate of CCN concentration (Doherty et al., 2022). Other factors causing differences in magnitude of CCN concentration and smoke plume placement between our lidar-derived CCN concentrations and WRF-CAM5 CCN concentrations are outside the scope of this study but remain an area of future research. Rather, we use this example flight path as an illustration of a way that the lidar-derived CCN can be used to evaluate and potentially improve model performance moving forward.

Lastly, we expand our data collocation criteria to analyze the applicability of our regression equations to a larger subset of the data collected in ORACLES. In Figure 9, the relationship between lidar-derived CCN concentration (using the regression equation for backscatter at 532 nm from Figure 3) and in situ CCN concentration is given. This expanded collocated data set of 1037 points is attained using a time separation criterion of ±0.5 hr, a horizontal distance criterion of ±4.4 km, and a vertical bin size of 45 m. Our sensitivity testing revealed that these criteria result in more scatter and noise

within the in situ CCN – HSRL-2 observable linear relationship, so while they were not used to derive the regression equations, they are used here solely to test our regression method on a larger subset of the data across September 2016, August 2017, and September/October 2018. A correlation coefficient of 0.87 for this expanded data set shows the general applicability of our regression equations to a significantly larger data set. Comparing this to a correlation coefficient of 0.97 for the original data set also supports our original decision to limit the collocation criteria to relatively small spatiotemporal

extents. Similarly, comparing points considered in the original data set (outlined in black) to those from the expanded data set show the redundancy that occurs when expanding our horizontal criteria, where in situ observations are collocated with and compared to multiple lidar observations. Therefore, while our original data collocation criteria served to optimize the strength of these linear relationships, the expanded criteria are used to show the applicability of the method to a larger portion of the data.





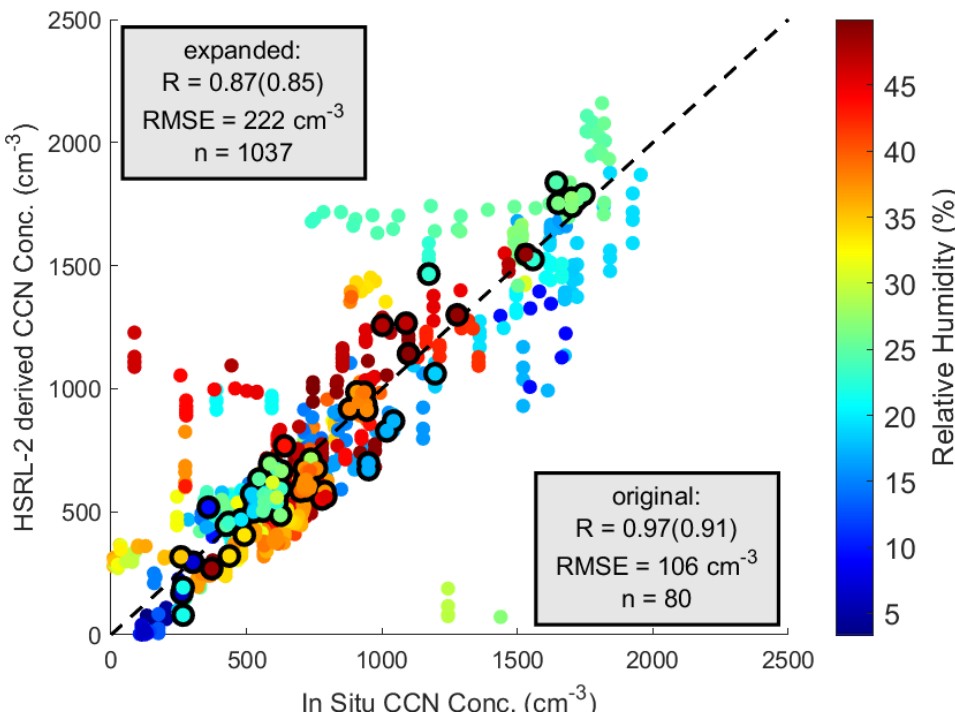


**Figure 9: In situ CCN concentration vs. HSRL-2 derived CCN concentration is given for an expanded three year data set (dt = ±.0.5hr, dd = ±4.4km, dh = 45m). The supersaturation in the expanded data set ranges from 0.15-0.4% and the supersaturation for the original collocated data set ranges from 0.22-0.4%. The points with a black outline**
**designate those included in the original collocated data set that was used to develop the lidar-derived CCN concentration. The dashed line represents the 1:1 line, and for each data set the Pearson correlation coefficient is shown, with Spearman rank correlation coefficient given in parentheses. This comparison shows the general applicability of our regression equations to derive CCN concentrations using HSRL-2 observables over a larger subset of the ORACLES observations.**

The relationships and regression equations used to derive CCN concentrations from HSRL-2 observables in this study are specific to BBA in the SEA. Since CCN activation depends strongly on aerosol size and chemical composition, their relationships to HSRL-2 observables will vary for different aerosol types. However, this same methodology of using collocated data to develop linear regressions between in situ CCN concentrations and HSRL-2 observables can be applied to allow for lidar-derived CCN concentrations for different aerosol types and locations.


**5 Conclusion**

To improve our understanding of aerosol-cloud interactions and aerosol-induced radiative effects, knowledge of aerosol concentrations, sizes, and spatial distributions are essential. Several studies have used remote sensing techniques to glean





such information. However, many studies rely on the use of column-effective products from passive remote sensing such as

AOD as a proxy for aerosol or CCN concentration. This approach requires significant assumptions and results in a lack of information about vertical distributions, which are critical in evaluating the aerosol indirect effect. In this study, we investigate correlations between in situ measured CCN concentrations and vertically-resolved HSRL-2 measurements in the ORACLES campaign, and we use them to derive vertically-resolved CCN concentrations of BBA.

        We find that CCN concentrations and HSRL-2 backscatter and extinction coefficients are positively and linearly

correlated over the SEA BBA-dominated region when observations are restricted to low RH. We find an optimum between data availability and correlation strength when averaging over a horizontal separation distance of 2 km, a temporal separation between observations of ± 6 to 18 minutes, and vertical bins of 45-75 m. Even with these strict data collocation constraints, we find that the collocated data set are sufficiently representative of the total data set.

        After data collocation, we analyze the relationships between in situ CCN concentrations and HSRL-2 backscatter

and extinction coefficients for all three years of ORACLES. When analyzed together, this combined data set results in correlation coefficients between 0.95-0.97 between CCN concentrations and HSRL-2 backscatter and extinction coefficients, respectively. In addition, there is no difference in correlation between 355 and 532 nm wavelengths for this combined data set. When using AI calculated from HSRL-2 extinction, similar relationships appear, with AI from extinction at 532 nm having a slightly stronger relationship with CCN concentration than the 355 nm counterpart. We do not find that AI was

significantly better at representing small CCN aerosols than extinction alone. One important caveat is that these relationships work well only for conditions with low ambient RH. For future analyses performed over a wider range of RH (e.g., estimating CCN concentrations in the marine boundary layer), AI may prove more beneficial.

        When looking at the representativeness and applications of this work, we find that observations from three years of ORACLES account for multiple different measurement locations and aerosol ages relative to the commonly occurring

trajectory of the seasonal biomass burning plume. While our collocated data set does not account for the entirety of the SEA, the range that is represented can be analyzed collectively in that the strong linear relationships between CCN concentrations and HSRL-2 observables are robust. Additionally, we find that there are minimal differences caused by sampling location and aerosol age. These two findings suggest that the collocated data set is well-representative of the BBA-dominated SEA. Lastly, we compare campaign-average values of CCN concentrations and in situ ("dry") extinction coefficients to results

from Shinozuka et al. (2015) and find that our BBA-dominant results from ORACLES are comparable to those from other regions with different dominant aerosol types. This finding suggests that a similar analysis using in situ CCN and HSRL-2 observations from other campaigns and regions could be feasible, allowing for an extension of the lidar-derived CCN concentrations to other locations and aerosol types. A case study for a specific HSRL-2 curtain in 2017 points to a few important differences between the lidar-derived and WRF-CAM5 modeled CCN concentrations.

Overall, these results support the plausibility and reproducibility of using HSRL-2 observables to quantitatively obtain CCN concentrations in BBA dominated air masses. In light of a potential future spaceborne HSRL, as outlined by the NASA Atmosphere Observing System (AOS) mission, it is highly beneficial to develop methods to enable future use of such



a system to address AOS goals related to improving information about vertically-resolved aerosol and CCN concentrations. For example, Stier et al. (2016) suggested that a spaceborne HSRL could advance observational constraints on CCN, and this
study acts as an attempt to study this constraint. More widely available and easily accessible CCN concentration data would aid in further studies of aerosol-cloud interactions, ultimately reducing the uncertainty of their contribution to aerosol radiative forcing of climate.

**Data Availability**

The ER-2 and P-3 data are available through the NASA data archive: https://espo.nasa.gov/oracles/archive/browse/oracles

**Author Contributions**

EDL, LG, and JR formulated the CCN and lidar observables study. EDL and LG organized all data products, performed analyses, and visualized the results. EDL wrote the draft. LG, JR, FX, SPB, BC, IC, RAF, PES, CH, YS, AD, SF, SS, and
AN edited the manuscript and provided insightful discussion and suggestions.

**Competing Interests**

The authors declare that they have no conflict of interest.

**Acknowledgements**

The team acknowledges the contributions of the entire NASA ORACLES team and funding from the Earth Venture Suborbital-2 (EVS-2) program (grant no. 13-EVS2-13-0028). Part of the computation in this paper was performed at the Supercomputing Center for Education & Research (OSCER) at the University of Oklahoma (OU). PES and CH acknowledge support from Department of Energy grant DE-SC0018272. AN and MK acknowledge support by NASA
ORACLES. AN acknowledges support from project PyroTRACH (ERC-2016-COG) funded from H2020-EU.1.1.—Excellent Science—European Research Council (ERC), project ID 726165.

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
