# Peer review of "Use of Lidar Aerosol Extinction and Backscatter Coefficients to Estimate Cloud Condensation Nuclei (CCN) Concentrations in the Southeast Atlantic"

_Atmospheric Measurement Techniques, 2022_

## Author Comment (AC2)

We would like to thank the reviewer for the following constructive feedback on our manuscript and in aiding our progress towards publication. Our responses are given in blue text and any adjustments to the manuscript text are given in quotes and italics. For multi-part comments the specific part of the comment being addressed is given again in bold **[brackets]** before the corresponding response. Major changes to the updated manuscript text are highlighted.

This paper uses a recent observational dataset to examine the relationships between lidar measurements (backscatter and extinction) and inlet-based in situ CCN concentrations, with the ultimate goal of evaluating model performance and HSRL-based CCN retrievals. While the paper is well-written, I worry the conclusions are somewhat simplistic and not fully supported by the data or uncertainty analysis as presented. I would like revision to clarify the below questions before the paper is published.

**Major comments:**

1) While there's nothing inherently wrong about a straightforward approach, the basic methodology as I understand it (essentially, using a linear fit to estimate CCN concentration from HSRL data) merits a more detailed description. For example, which "best fit" lines are shown in the figures? I didn't see it described. Many standard statistics packages use ordinary least-squares, which presumes that the x-variable is a perfect measurement and all the error/uncertainty is in the y-variable, but for a regression between two observed variables, I don't think that's accurate. Surely HSRL-2 comes with uncertainty as well? How was this accounted for? It also seems like OLS may have been used since Figs 3 and 4 and 9 show RMSE, in units of CCN concentration, i.e. the y-variable. (Follow-up question: what's the utility of this metric here? Is it to say that the CCN uncertainty associated with the linear regression is ~>100/cm3 in each case? Surely there's more to it than just that? How does an estimate of uncertainty in the linear regression influence the later results, e.g. Fig 9, 8?)

**[While there's nothing inherently wrong about a straightforward approach, the basic methodology as I understand it (essentially, using a linear fit to estimate CCN concentration from HSRL data) merits a more detailed description. For example, which "best fit" lines are shown in the figures? I didn't see it described. Many standard statistics packages use ordinary least-squares, which presumes that the x-variable is a perfect measurement and all the error/uncertainty is in the y-variable, but for a regression between two observed variables, I don't think that's accurate.]**

We acknowledge the discrepancy between our original (ordinary least squares) regression method and the fact that both variables here are measured with error. We have changed our approach to use a bisector regression, as shown in Figures 3, 4, and 6. The bisector regression bisects the angle between y vs. x and x vs. y regressions and weights measurements of both variables with their error. This change is described in Lines 324-325 as follows:

*"All relationships are fit using a bisector regression to account for both variables being measured with uncertainty."*

**[It also seems like OLS may have been used since Figs 3 and 4 and 9 show RMSE, in units of CCN concentration, i.e. the y-variable. (Follow-up question: what's the utility of this metric here? Is it to say that the CCN uncertainty associated with the linear regression is ~>100/cm3 in each case? Surely there's more to it than just that?]**

In addition, we now include vertical and horizontal error bars to indicate measurement uncertainty (more about HSRL-2 uncertainty below) as well as a relative slope uncertainty for each regression to account for uncertainty associated with the regression itself. In addition, we agree that RMSE is not equivalent to derived CCN uncertainty. Rather, it was included as one of the goodness of fit metrics to represent scatter around the fit. RMSE was described and compared in the writing to the magnitude of in situ CCN uncertainty to relate the two but not treat them as the same thing. These changes are described in the text between Lines 326-336 as follows:

*"We show the Pearson correlation coefficient (R), Spearman rank correlation coefficient (in parentheses), root mean square error (RMSE), percentage of data within ±10% of the linear regression line, and relative uncertainty of the slope. In addition, we plot error bars representing relative CCN uncertainty (vertical) and calculated HSRL-2 uncertainties (horizontal). The combination of data from different measurement periods across three years results in strong correlations (0.95-0.97) between both variables. This result suggests that our collocation and filtering methodology is reasonable and holds well for multiple observational periods. In addition, different symbols designating different years support the representativeness of the collocated data set, as no one period of observations completely stands out from another. RMSE is on the order of 100 cm$^{-3}$. With a relative CCN uncertainty of 10%, RMSE is of the same order of magnitude as a median CCN uncertainty for this data set. The amount of data within ±10% of each respective linear regression line ranges from 55-66%. These values, together with RMSE, suggest a relatively low amount of scatter around the regression line for both wavelengths and coefficients."*

**[Surely HSRL-2 comes with uncertainty as well? How was this accounted for?]**

HSRL-2 uncertainty was not accounted for in the original regression, as officially reported HSRL-2 uncertainty values are only available for ORACLES 2016, because the instrument was located on a high-flying platform in this campaign, allowing for careful calibration with clear-air returns. However, upon receiving this question we reached out to the ORACLES HSRL-2 team and were advised on how to calculate approximate uncertainty values for 2017 and 2018 data. This was done using a spatial variability method that we describe in a new Section 2.3 titled "HSRL-2 Uncertainty Calculations" (lines 291-310). We take HSRL-2 observables from 5 profiles before and 5 profiles after each collocated data point (at the same altitude) and calculate the standard deviation across all profiles. This results in one HSRL-2 uncertainty value for each collocated data point.

**[How does an estimate of uncertainty in the linear regression influence the later results, e.g. Fig 9, 8?]**

We then compare our uncertainties calculated using this spatial variability method with the reported uncertainties available for September 2016 (Table 3). Through this comparison we show that our mean calculated uncertainties are on the same order of magnitude as officially reported uncertainties. However, our calculated values do span a wider range than reported uncertainties, suggesting that this method captures a possible upper-bound to HSRL-2 uncertainties. Since the means from our method match well with reported mean values and the ranges do not tend to underestimate uncertainty, we move forward with using our calculated values to depict HSRL-2 error via horizontal error bars on Figures 3 and 6. A plot is provided for the reader below to visualize the comparison between reported and calculated uncertainties. Section 2.3 and Table 3 read as follows:

*"The forthcoming analysis develops a regression between HSRL-2 observables and CCN concentration, both of which are observed quantities measured with uncertainty. Therefore, we consider uncertainties associated with both measurements and with the slope of each regression. At the time of this analysis, reported HSRL-2 uncertainties were only available for September 2016. Therefore, we have taken a spatial variability approach to estimate uncertainties for HSRL-2 data from August 2017 and September-October 2018. This method uses backscatter and extinction coefficients in the same vertical bin from five profiles before and five profiles after the HSRL-2 profile associated with each collocated data point. We analyze the distributions of backscatter and extinction across these profiles to ensure no large variations in either coefficient, i.e., that we are accurately estimating instrument uncertainty and not including a large gradient due to aerosol spatial inhomogeneity. After this step we calculate the standard deviation across all eleven profiles to use as a measure of uncertainty.*

*In Table 3 we present a comparison between HSRL-2 uncertainties calculated using this spatial variability method to the reported HSRL-2 uncertainties available for September 2016. In general, the mean uncertainties from both methods are on the same order of magnitude and very close in value. However, our calculated uncertainties span a wider range than reported uncertainties, suggesting that this method captures a possible upper-bound to HSRL-2 uncertainties. While this method only accounts for random uncertainties in backscatter and extinction measurements, systematic uncertainty for backscatter is reported as 5% for 355 nm and 4.1% for 532 nm while extinction is dominated by random error and has a small systematic error (Burton et al., 2015). Given the similar mean uncertainties and possible slight overestimation of HSRL-2 reported uncertainty (rather than consistent underestimation of error) using our spatial variability method, we use these values when considering uncertainty impacting the forthcoming regressions. Furthermore, we present this spatial variability method as a reasonable way to estimate HSRL-2 uncertainties in future studies."*

**Table 3: Comparison of HSRL-2 reported uncertainty to uncertainties calculated using a spatial variability method for September 2016.**

| | Reported HSRL-2 Uncertainty | | HSRL-2 Uncertainty Calculated from Spatial Variability Method | |
|---|---|---|---|---|
| | Range | Mean | Range | Mean |
| BSC355 ($km^{-1}sr^{-1}$) | 5.5E-05 – 2.1E-04 | 1.5E-04 | 6.3E-05 - 4.17E-04 | 1.4E-04 |
| BSC532 ($km^{-1}sr^{-1}$) | 3.2E-05 – 7.3E-05 | 6.2E-05 | 2.9E-05 - 2.1E-04 | 7.0E-05 |

| EXT355 (km$^{-1}$) | 4.7E-03 – 1.2E-02 | 8.8E-03 | 2.1E-03 – 1.5E-02 | 7.6E-03 |
|---|---|---|---|---|
| EXT532 (km$^{-1}$) | 3.8E-03 – 5.6E-03 | 4.9E-03 | 1.9E-03 – 1.7E-02 | 6.4E-03 |

Figures 3, 4, and 6 are updated to include error bars in the manuscript and are shown below. Their captions have also been edited to reflect changes made to the plots. Note that Figure 4 does not include horizontal error bars. This is due to the fact that extinction and Angstrom exponent are not independent of each other, there is no straight-forward way to calculate the propagation without estimates of error covariance, and the uncertainties in aerosol index are not further used in the manuscript.

[Figure]

[Figure]

[Figure]

While we now consider HSRL-2 and CCN uncertainty using bisector regression, error bars, and a measure of slope uncertainty, we do not propagate all sources of uncertainty into one value to be used as an estimate of uncertainty associated with lidar-derived CCN concentration. This is again because not all sources of error within these regressions are independent of each other – error in the slope is dependent on both HSRL-2 and CCN uncertainty. That is, the assumptions of error propagation do not hold when trying to incorporate each of these sources of error into one value. Therefore, we are do not calculate a propagated error value applicable to our lidar-derived CCN concentrations. However, we acknowledge that we use a simplified methodology with multiple possible sources of uncertainty. We discuss all sources of uncertainty in a new Section 4.3 titled "Sources of Uncertainty" that aims to qualify the performance of these regressions in light of the aforementioned sources of uncertainty. Section 4.3 reads as follows:

*"As previously mentioned and taken into consideration via bisector regression, both CCN and HSRL-2 are observations made with uncertainty. Relative CCN uncertainty is 10% and our spatial variability method of calculating HSRL-2 uncertainties resulted in mean values of 1.4E-04 km$^{-1}$sr$^{-1}$ and 7.0-E-05 km$^{-1}$sr$^{-1}$ for backscatter at 355 and 532 nm, respectively, and 0.0076 km$^{-1}$ and 0.0064 km$^{-1}$ for extinction at 355 and 532 nm, respectively (Table 3). In addition, uncertainty is introduced as a result of the regression itself. Relative slope uncertainties range from 3.0-3.6% (Figure 3). We discuss each of these sources of error separately due to their dependence on one another. Uncertainties in both CCN concentration and HSRL-2 observations will impact uncertainty in the slope of each regression. Therefore, the assumption of each source of error being independent that is required for error propagation calculations does not hold. Rather, we present our method of deriving CCN concentration from lidar observables with such explanation of the various sources of error that will impact results.*

*In addition to observational and regression-based uncertainties, another possible source of error when applying this method stems from the specific characteristics of the data set used to develop the regression equations. The relationships analyzed in this study are specific to BBA in the SEA. Additionally, they are specific to ambient conditions with low RH (≤40-50%), S≥0.2%, and aerosol ages represented by f44 values between about 0.17-0.27. While these conditions are characteristic of the high-altitude SEA smoke plume, they will not hold in all regions and for all aerosol types. Therefore, without careful consideration of the ambient conditions and aerosol types to which the regressions derived here are applied, increased uncertainty will be introduced in lidar-derived CCN concentrations. Despite the strict conditions under which our regressions are applicable, we will explore their performance on a larger portion of the collocated data set in the following section."*

1a) Specifically to the above: Line 360-361 mentions "higher uncertainties and coarser resolutions associated with the [HSRL-2] extinction coefficient"-- where is this incorporated or considered?

As mentioned above, we initially did not have uncertainty values for the entire HSRL-2 data set. Therefore, this statement about higher uncertainties for extinction was meant to speak to our interpretation of the differences in regression strength between backscatter and extinction. Though extinction does have a coarser native resolution when compared to backscatter, observations are adjusted in the data set to match backscatter resolutions. This sentence has been adjusted slightly to avoid the confusion of different resolutions and to communicate that the differences in regression strength could (qualitatively) be different due to higher extinction uncertainty. The adjusted sentence reads as follows (lines 389-391):

*"The relationship between in situ CCN concentrations and HSRL-2 backscatter coefficients is slightly stronger than that using extinction coefficients, which could be due to higher uncertainties associated with the extinction coefficient (Burton et al., 2016)."*

2) The authors also combine all three years (with different measurement times of each) into one plot, which could be fine, but in Figs 3, 4, and 6, it seems the goodness of the fit is likely strongly influenced by those cluster of high CCN, high X (X= the given HSRL-2 variable) which occur specifically in 2017. Plus the choice to force through 0-0. There are only 13 points in 2017, all >~900cm-3, which is almost an entirely different range than 2016, for example. Again, this probably could be fine, *if* the CCN/HSRL relationship(s) hold over the range of conditions as the months change. The authors consider different metrics f44, AI (if not AE) (mostly in Section 4.2), and changes in SS level, RH %, and measurement constraints for different years (Table 2), but with so few points from so few days, it's hard to determine how consistent this relationship actually is in a multivariate sense, just from what has been presented.

**[The authors also combine all three years (with different measurement times of each) into one plot, which could be fine, but in Figs 3, 4, and 6, it seems the goodness of the fit is likely strongly influenced by those cluster of high CCN, high X (X= the given HSRL-2 variable) which occur specifically in 2017. Plus the choice to force through 0-0. There are only 13 points in 2017, all >~900cm-3, which is almost an entirely different range than 2016, for example.]**

Though we did not show results separated by year in the paper, this was a step of our analysis as we collocated and filtered (RH, supersaturation) the data one year at a time. During those steps the fit lines appeared very similar in terms of slope, which motivated the decision to combine all three years of data. However, the importance of ensuring that the combined goodness of fit is not fully dominated by the cluster of data from 2017 is well-taken. Therefore, the plots below give an example of bisector regressions for CCN vs. backscatter and extinction at 355nm with the three years separated.

[Figure]

Here we acknowledge that the fits between all three years are slightly different. Though they are not identical, the slopes are similar in magnitude and span a relatively small range. The reviewer pointed out that the goodness of fit appeared to be largely influenced by the high CCN data points from 2017. Though correlation coefficient is dependent on the range of the data, here we see similar R values for each individual year as we see when all years are

combined. In addition, RMSE values do not vary significantly from year to year. Therefore, while 2017 does have a larger y-intercept the similarities in other characteristics of the fit (slope, R, RMSE) motivated us to still move forward with combining all three years of data. To further address our reasoning for combining all three years, the text in lines 321-324 was adjusted to read as follows:

*"After analyzing year-by-year relationships and finding similar fit line slopes for any given HSRL-2 coefficient and wavelength, we consider all collocated data points in Figure 3. Data from September 2016, August 2017, and September-October 2018 are combined to fit a relationship between CCN concentration and HSRL-2 backscatter and extinction at 355 and 532 nm."*

**[Again, this probably could be fine, \*if\* the CCN/HSRL relationship(s) hold over the range of conditions as the months change. The authors consider different metrics f44, AI (if not AE) (mostly in Section 4.2), and changes in SS level, RH %, and measurement constraints for different years (Table 2), but with so few points from so few days, it's hard to determine how consistent this relationship actually is in a multivariate sense, just from what has been presented.]**

The reviewer also asked about how consistent the relationship is in a multivariate sense. We acknowledge that the constraints that we have put on the data are rather strict and will not hold for all scenarios. These regressions should not be applied in all scenarios. However, we have tried to express that the regression equations are meant to be specific to biomass burning aerosols in the Southeast Atlantic smoke plume, which is primarily found at altitudes where ambient RH is often dry (< 40-50%) and for higher end supersaturations where aerosols are more likely to activate as CCN. In this study we developed a regression very specific to smoke aerosols in this environment, and our goal was not for the relationships to hold over a wide variety of conditions. In the future when using data from other regions and campaigns our goal is to look at relationships for multiple aerosol types and a wider range of ambient RH. However, to clarify the scope of our regressions in this study, the following sentences have been added between lines 499-501:

*"Additionally, they are specific to ambient conditions with low RH (≤40-50%), S≥0.2%, and aerosol ages represented by f44 values between about 0.17-0.27. While these conditions are characteristic of the high-altitude SEA smoke plume, they will not hold in all regions and for all aerosol types."*

We would like to note here that it is very difficult to get collocated data such as we have here, even in a field mission dedicated to it. It may be a drawback that this study has a bit of a preliminary/exploratory style, but this speaks to the motivation for wanting to do this study in the first place – our goal is being able to infer CCN from more widespread measurements. With that being said, we acknowledge that the collocated data set was slightly limited in 2017/2018 (compared to 2016) due to the HSRL-2 and CCN counter being located on the same aircraft and operating at different times. In addition, our collocation criteria were intentionally set up to optimize the strength of the relationships though it limited the amount of collocated data points. Figure 8 (originally labelled as Figure 9) speaks to the

dilemma we faced, where expanding our criteria to include more data points resulted in weaker relationships that would introduce more error into our lidar-derived CCN concentrations.

**2a)** Further comments on Fig 6: am I reading it right that the thicker grey line is for 2018? The three diamond points I can see, do not appear to fit well to that line, since the two higher points are both below that fit and other two are right on it. Is that correct? How can that be the best fit to those four points? And are there really only two data points from 2017? Is that enough to draw conclusions from?

**[Further comments on Fig 6: am I reading it right that the thicker grey line is for 2018? The three diamond points I can see, do not appear to fit well to that line, since the two higher points are both below that fit and other two are right on it. Is that correct? How can that be the best fit to those four points?]**

We would like to thank the reviewer for catching this issue with Figure 6! The original Fig. 6 included only the scatter points that had a f44 value available but used the regression lines from the entire year-by-year collocated data set fit. This is the reason that the three diamond points did not seem to fit well to the 2018 fit line. This plot has been updated to include the entire collocated data set, and the points that are filled with gray have no f44 value available. In addition, this figure was updated to include only the three-year combined regression line (the same as shown in Figure 3). This was done to avoid similar confusion as to whether each fit line was specific to only those points with f44 values available – we felt that the point of the plot was more straightforward this way and better corresponded with how we described Figure 6 in the text. The updated figure is shown in response to comment 1.

**[And are there really only two data points from 2017? Is that enough to draw conclusions from?]**

In terms of the amount of missing data for 2017 specifically (though all years are missing several f44 values), the intention with this plot was not to draw conclusions based on the fit between CCN and HSRL-2 for points with an f44 value. This might have been misconstrued based on the confusion between scatter points and regression lines in the original plot. Additionally, as the figure is updated now the purpose remains the same – we are aiming to communicate that our collocated data set covers a fairly wide range of aerosol ages and that no end of this range causes the regression to break down. Therefore, we are confident that this fit should apply well for a range of aerosol ages in the Southeast Atlantic. To further clarify this point, the following sentence has been added to the end of the 2nd paragraph in Section 4.2 (lines 439-442):

*"Furthermore, though a complete f44 data set is not available for every point in our collocated data set, the values that are available suggest that we consider a fairly wide*

*range of aerosol ages in this analysis with no singular age breaking down the strength of the regression (Figure 6)."*

The regression line symbols were adjusted to make this plot easier to read after adding the gray scatter points. The caption was also adjusted to reflect this change.

**2b)** Finally, I worry that the methodology of fitting through 0, while physically intuitive, is constraining the results in a way that's not supported by the (somewhat limited) data which are shown. To take an extreme example, if one fits a line through (0,0) and 2 other data points which have some error associated, the fit will likely be hugely different compared to a fit through just those two data points themselves. The artificial (0,0) "data" would completely overwhelm the relationship between the actual observed datapoints, which is what the authors are trying to show. And obviously this would have a more dramatic effect for studies with fewer datapoints.  How much does this fit depend on the (0,0) constraint? It seems it may have a big effect here.

Our decision to fit the regression lines through (0,0) was primarily motivated by considering the relationship between optics and CCN concentration theoretically. Theoretically, it is not conceivable to have CCN present with absolutely no backscatter or extinction return. Similarly, it isn't conceivable to have some backscatter or extinction return for a size distribution with no CCN. However, the reviewer makes a good point that it is not obvious how this decision impacts the fit lines. Below is an example of a plot with fit lines not forced through (0,0) to compare with Figure 3 in the manuscript.

[Figure]

When not fitting through (0,0) the slope for the backscatter 532nm regression line is 6.49E05, compared to a value of 6.96E05 when fitting through (0,0). When not fitting through (0,0) the slope for the backscatter 355nm regression line is 3.20E05, compared to a value of 3.50E05 when fitting through (0,0). When not fitting through (0,0) the slope for the

extinction 532nm regression line is 9.07E03, compared to a value of 9.86E03 when fitting through (0,0). When not fitting through (0,0) the slope for the extinction 355nm regression line is 5.25E03, compared to a value of 5.70E03 when fitting through (0,0). In addition, the relative slope uncertainties for each method vary only by 0.1-0.4%. Therefore, we acknowledge that while fitting through (0,0) does have an impact on the fit that it results in a very small difference in slope and slope uncertainty such that the theoretical reasoning behind this choice is acceptable. Furthermore, the y-intercepts resultant from not fitting through (0,0) are also relatively small (79.6-138 cm$^{-3}$) and relatively close to (0,0), especially in the context of the spread of CCN concentration values on the y-axis.

**3)** I'm not clear on the purpose of Sec 4.3/Fig 8-9. First, is (or can) WRF-CAM5 be taken as a ground truth, or can HSRL-2? (I suspect neither, with the information presented). The two plots in Fig 8 are so different I'm having trouble understanding what's the message here. The authors (Line 464) suggest that the lidar-derived method is "better" than the models. Based on what?

The reviewer's point that we haven't proven either WRF-CAM5 or HSRL-2 as able to be taken as ground truth is well taken. Furthermore, we were unable to do a full quantitative analysis of the differences in CCN as represented by our lidar-derived method and what is given in WRF-CAM5 due to a complete WRF-CAM5 data set for ORACLES not yet being available. The main point of this figure was to show our produced data set in context of one model result to show similarities and differences, as more of a forward-looking aspect of the analysis. Additionally, the use of "better" attempted to address the differences in detail and higher resolution provided by our lidar-derived method when compared to the model output, though that was not directly stated in the original version. Therefore, the order of Fig 8 and 9 is switched (further addressed below) and the wording of lines 548-551 has been changed to the following:

*"However, our lidar-derived method results in a better estimate of smoke plume depth and altitude, cloud top heights, and small-scale variation of CCN concentrations. This is likely due to our lidar-derived method having a higher horizontal resolution (2 km from HSRL-2 compared to 36 km from WRF-CAM5), as well as the depiction of possible turbulence and entrainment effects that are not currently captured by the model."*

**3a)** Figure 9 has many points which suggest variability in in situ CCN which is not captured by the HSRL-2-derived product, as well as some which suggest artificial variability in HSRL-2 compared with in situ. Some cases seem to show a mismatch of an order of magnitude. What's going on in these cases? Is this solely a function of the "expanded" dataset, i.e. those strong diversions are a result of greater mismatches in space/time? Does it have anything to do with the age/humidity/supersaturation? How does this fit into Fig 5, which at least for 2017 seems to show both young and old aerosol? Does this matter? As presented here it's difficult to believe in the results of Fig 9.

**[Figure 9 has many points which suggest variability in in situ CCN which is not captured by the HSRL-2-derived product, as well as some which suggest artificial variability in HSRL-2 compared with in situ. Some cases seem to show a mismatch of an order of magnitude. What's going on in these cases? Is this solely a function of the "expanded" dataset, i.e. those strong diversions are a result of greater mismatches in space/time?]**

The original version of Figure 9 looked at an expanded data set using larger time and horizontal criteria by which CCN and HSRL-2 data sets are collocated. However, as the reviewer points out, some of these points seem to suggest artificial variability in HSRL-2 compared with in situ. This was likely due to increasing two criteria at once - for sequential HSRL-2 profiles, larger time and horizontal criteria result in some overlap in in situ CCN being averaged to relate to an average backscatter or extinction value. Therefore, some of the scatter in our original Figure 9 was not communicating what we initially intended. Figure 9 has been replaced by an updated version (below) that only expands the horizontal criteria to ±4.4km (from an original ±1.1km). These changes are updated in the figure caption as well. This reduces the amount of artificial scatter and allows us to focus on the impact of spatial variability and how it impacts the correlation.

[Figure]

**[Does it have anything to do with the age/humidity/supersaturation? How does this fit into Fig 5, which at least for 2017 seems to show both young and old aerosol? Does this matter? As presented here it's difficult to believe in the results of Fig 9.]**

Here, we see an increase in scatter around the one-to-one line when using an expanded data set, which speaks to two main points. First, our original tighter criteria allow us to hone in on a tighter relationship between CCN and HSRL-2 coefficients, as scatter increases when we introduce more spatial variability. Secondly, this method is not overly specialized to the point where it fails when being tested on a larger portion of the data. The humidity range is the same between Figure 9 and our other figures. The supersaturation range only varies slightly because 2018 uses a supersaturation filtering threshold of $S \geq 0.4\%$, thus allowing for 9 points with $S < 0.2\%$. Below is a figure showing the distribution of f44 values represented by this expanded data set. Though there are a few lower values than were represented in Figure 5, most values fall in the same range of 0.18-0.28, indicating that any scatter in Figure 9 is not due to completely different ages of aerosols.

[Figure]

The paragraph from lines 508-522 was adjusted in the following way:

*"In addition to showing the representativeness of our lidar-derived CCN method in the context of additional aerosol types and data sets, we expand our data collocation criteria to analyze the applicability of our regression equations to a larger subset of the data collected in ORACLES. In Figure 9, the relationship between lidar-derived CCN concentration (using the regression equation for backscatter at 532 nm from Figure 3) and in situ CCN concentration is shown. This expanded collocated data set of 460 points is attained using by increasing the horizontal distance criterion to ±4.4 km for each year, leaving vertical bin size and time criteria the same (Table 2). Our sensitivity testing revealed that using a larger horizontal distance results in more scatter and noise within the in situ CCN – HSRL-2 observable linear relationship, so while this value was not used to derive the regression equations, it is used here solely to test our*

*regression method on a larger subset of the data across September 2016, August 2017, and September/October 2018. A correlation coefficient of 0.85 for this expanded data set shows the general applicability of our regression equations to a larger data set, demonstrating that our regression equations are not over-fitted to the smaller original set of collocated data points. Comparing this to a correlation coefficient of 0.97 for the original data set also supports our original decision to limit the collocation criteria to relatively small spatiotemporal extents when developing the regressions to avoid the effects of larger spatial variability in CCN concentrations. Therefore, while our original data collocation criteria served to optimize the strength of these linear relationships, the expanded criteria are used to show the applicability of the method to a larger portion of the data."*

**3b)** If the two above points are addressed, I'd suggest flipping Figs 8 and 9; 9 should establish the validity of the HSRL-derived CCN product and then 8 could show it in the context of one model result. Perhaps that's more what the authors are intending.

Figures 8 and 9 are flipped in the updated draft to make the point that the reviewer mentions. Wording is slightly different to account for switching the order and smoothly transitioning between these two different points.

**4)** Figure 5, Lines 396-8: this is not correct. The main source of the BBA in this region is the south african easterly jet (AEJ-S) which is most frequently present between 5-15S, not at the equator. See: Adebiyi et al 2016 DOI:10.1002/qj.2765; Ryoo et al 2021 https://doi.org/10.5194/acp-21-16689-2021

We would like to thank the reviewer for pointing out this incorrect statement in Section 4.2. This sentence has been changed (lines 425-427) and the two references have been added as follows:

*"One commonly occurring trajectory for BBA in this region suggests that the plume core is lofted over the SEA between 5-15°S, whereby the aerosols move westward, following a spiral path, and later descend into the boundary layer near 20°S and close to the coast (Adebiyi & Zuidema, 2016; Redemann et al., 2021; Ryoo et al., 2021)."*

**Minor comments:**

For the "amount of data within +/- X% of the linear regression line," X=10% in the figure captions (Figs 3, 4) and 20% in the text (lines 300, 327, 330).

The correct value (used in the plots) is 10%, so this was fixed in the text.

The authors consider both extinction and AI and draw conclusions that the latter isn't more representative because AE has minimal variation (paragraph on Line 365), but presumably the AE used to calculate AI was also determined from HSRL-2 measurements. Did you

examine the AE variability directly to support this conclusion? It seems fairly straightforward to check  (I'm considering this a "minor comment" because this is more of a curiosity rather than a major issue with the paper).

Yes, the AE used to calculate AI was determined from HSRL-2 measurements. Below is a histogram depicting AE variability within the collocated data set, showing values that center around approximately 1-1.5.

[Figure]

Line 136: a plausible and robust collocation and filtering is central and critical to the subsequent results; in other words, I'd remove the word "briefly" here.

The word "briefly" was removed in line 136.

Line 147,224: Fig 1 caption says the third deployment was only Oct 2018?

Thank you for catching this discrepancy! The Figure 1 caption was updated, and the plot was also slightly changed to one whose colors match Redemann et al. (2021).

Line 170: "the exact temporal" … resolution?

The word "resolution" was added in line 170.

Line 191: is this — here necessary? It seems a bit awkward. Maybe needs a comma?

The dash was changed to a comma in line 191.

Line 196: how do +/- 10% and 5-10 cm-3 compare to one another in absolute terms?

These values will compare to each other differently depending on magnitude of CCN concentration and amount of data measured at high S/N ratio vs. low S/N ratio. Since an uncertainty value of 10% was used when plotting error bars, the 5-10 cm-3 part was taken out of this sentence for clarity.

Line 200-1: revise, I don't follow. Maybe missing the word "fraction"?

Lines 200-201 were revised to be clearer.

Figure 2b: this uses both \deltat and dt in different places (also line 231, 235)– should this be the same notation?

Here, we used dt to represent the time criteria used to collocated both data sets. The Δt was used to represent the longer time gap between observations that we had to consider for 2017 and 2018 when both the HSRL-2 and CCN counter were located on the same aircraft. To make this distinction clearer, the Figure 2 caption has been adjusted to read as:

*"Figure 2: Graphic depicting data collocation process for a) 2016 and b) 2017-2018. CCN concentration measurements that fall within the time (dt), horizontal distance (dd), and vertical bin (dh) criteria (green points) are averaged to compare to the average HSRL-2 backscatter and extinction coefficients that fall within the same vertical bin. In b) t and Δt represent the longer time difference between measurements that must be considered for the 2017 and 2018 aircraft set-ups."*

Figure 3a: the legend covers the 532nm datapoints; resize or shift

Figure 3 was adjusted so that no points are covered by the legend.

---

## Author Comment (AC3)

We would like to thank the reviewer for the following constructive feedback on our manuscript and in aiding our progress towards publication. Our responses are given in blue text and any adjustments to the manuscript text are given in quotes and italics. Major changes to the updated manuscript text are highlighted.

Review of "Use of Lidar Aerosol Extinction and Backscatter Coefficients to Estimate Cloud Condensation Nuclei (CCN) Concentrations in the Southeast Atlantic" by Lenhardt et al., submitted to Atmospheric Measurement Techniques, 2022.

Overview:

This paper presents empirical relationships between remote sensing and in situ measurements of aerosol properties that were made during the NASA ORACLES project. The goal is to inform vertically-resolved CCN concentration retrieval algorithms that are heavily based on HRSL-2 data in the southeastern Atlantic airmasses dominated by smoke. The results presented in the form of correlation coefficients indicate that there is a strong relationship between HSRL-2 observations and the in situ CCN measurements from aircraft mounted sensors.

Review:

The paper is well organized and written. The figures complement the conclusions and are laid out appropriately. I do not find the conclusions to be overwrought because the authors state that the correlations described are limited to the SEA region and BBA type that was observed during ORACLES. However, there is a general reliance on the HSRL-2 observations without adequate caution. The authors are experienced with this system, so I recommend they include a more complete description of the limitations of the instrument on the airborne platform and the how the error propagates into the relationships derived herein, especially with regards to volume averaging extinction and backscatter coefficients. After the inclusion of such a discussion, I would find the paper suitable for publication.

We appreciate the kind feedback on the positive aspects of our paper. HSRL-2 uncertainty was not accounted for in the original regression, as officially reported HSRL-2 uncertainty values are only available for ORACLES 2016, because the instrument was located on a high-flying platform in this campaign, allowing for careful calibration with clear-air returns. However, upon receiving this question we reached out to the ORACLES HSRL-2 team and were advised on how to calculate approximate uncertainty values for 2017 and 2018 data. This was done using a spatial variability method that we describe in a new Section 2.3 titled "HSRL-2 Uncertainty Calculations" (lines 291-310). We take HSRL-2 observables from 5 profiles before and 5 profiles after each collocated data point (at the same altitude) and calculate the standard deviation across all profiles. This results in one HSRL-2 uncertainty value for each collocated data point.

We then compare our uncertainties calculated using this spatial variability method with the reported uncertainties available for September 2016 (Table 3). Through this comparison we show that our mean calculated uncertainties are on the same order of magnitude as officially reported uncertainties. However, our calculated values do span a wider range than

reported uncertainties, suggesting that this method captures a possible upper-bound to HSRL-2 uncertainties. Since the means from our method match well with reported mean values and the ranges do not tend to underestimate uncertainty, we move forward with using our calculated values to depict HSRL-2 error via horizontal error bars on Figures 3 and 6. A plot is provided for the reader below to visualize the comparison between reported and calculated uncertainties. Section 2.3 and Table 3 read as follows:

*"The forthcoming analysis develops a regression between HSRL-2 observables and CCN concentration, both of which are observed quantities measured with uncertainty. Therefore, we consider uncertainties associated with both measurements and with the slope of each regression. At the time of this analysis, reported HSRL-2 uncertainties were only available for September 2016. Therefore, we have taken a spatial variability approach to estimate uncertainties for HSRL-2 data from August 2017 and September-October 2018. This method uses backscatter and extinction coefficients in the same vertical bin from five profiles before and five profiles after the HSRL-2 profile associated with each collocated data point. We analyze the distributions of backscatter and extinction across these profiles to ensure no large variations in either coefficient, i.e., that we are accurately estimating instrument uncertainty and not including a large gradient due to aerosol spatial inhomogeneity. After this step we calculate the standard deviation across all eleven profiles to use as a measure of uncertainty.*

*In Table 3 we present a comparison between HSRL-2 uncertainties calculated using this spatial variability method to the reported HSRL-2 uncertainties available for September 2016. In general, the mean uncertainties from both methods are on the same order of magnitude and very close in value. However, our calculated uncertainties span a wider range than reported uncertainties, suggesting that this method captures a possible upper-bound to HSRL-2 uncertainties. While this method only accounts for random uncertainties in backscatter and extinction measurements, systematic uncertainty for backscatter is reported as 5% for 355 nm and 4.1% for 532 nm while extinction is dominated by random error and has a small systematic error (Burton et al., 2015). Given the similar mean uncertainties and possible slight overestimation of HSRL-2 reported uncertainty (rather than consistent underestimation of error) using our spatial variability method, we use these values when considering uncertainty impacting the forthcoming regressions. Furthermore, we present this spatial variability method as a reasonable way to estimate HSRL-2 uncertainties in future studies."*

**Table 3: Comparison of HSRL-2 reported uncertainty to uncertainties calculated using a spatial variability method for September 2016.**

| | Reported HSRL-2 Uncertainty | | HSRL-2 Uncertainty Calculated from Spatial Variability Method | |
|---|---|---|---|---|
| | Range | Mean | Range | Mean |
| BSC355 ($km^{-1}sr^{-1}$) | 5.5E-05 – 2.1E-04 | 1.5E-04 | 6.3E-05 - 4.17E-04 | 1.4E-04 |
| BSC532 ($km^{-1}sr^{-1}$) | 3.2E-05 – 7.3E-05 | 6.2E-05 | 2.9E-05 - 2.1E-04 | 7.0E-05 |
| EXT355 ($km^{-1}$) | 4.7E-03 – 1.2E-02 | 8.8E-03 | 2.1E-03 – 1.5E-02 | 7.6E-03 |
| EXT532 ($km^{-1}$) | 3.8E-03 – 5.6E-03 | 4.9E-03 | 1.9E-03 – 1.7E-02 | 6.4E-03 |

Figures 3, 4, and 6 are updated to include error bars in the manuscript and are shown below. Their captions have also been edited to reflect changes made to the plots. Note that

Figure 4 does not include horizontal error bars. This is due to the fact that extinction and Angstrom exponent are not independent of each other, there is no straight-forward way to calculate the propagation without estimates of error covariance, and the uncertainties in aerosol index are not further used in the manuscript.

[Figure]

[Figure]

[Figure]

While we now consider HSRL-2 and CCN uncertainty using bisector regression, error bars, and a measure of slope uncertainty, we do not propagate all sources of uncertainty into one value to be used as an estimate of uncertainty associated with lidar-derived CCN concentration. This is again because not all sources of error within these regressions are independent of each other – error in the slope is dependent on both HSRL-2 and CCN uncertainty. That is, the assumptions of error propagation do not hold when trying to incorporate each of these sources of error into one value. Therefore, we do not calculate a propagated error value applicable to our lidar-derived CCN concentrations. However, we acknowledge that we use a simplified methodology with multiple possible sources of uncertainty. We discuss all sources of uncertainty in a new Section 4.3 titled "Sources of Uncertainty" that aims to qualify the performance of these regressions in light of the aforementioned sources of uncertainty. Section 4.3 reads as follows:

*"As previously mentioned and taken into consideration via bisector regression, both CCN and HSRL-2 are observations made with uncertainty. Relative CCN uncertainty is 10% and our spatial variability method of calculating HSRL-2 uncertainties resulted in mean values of 1.4E-04 km$^{-1}$sr$^{-1}$ and 7.0-E-05 km$^{-1}$sr$^{-1}$ for backscatter at 355 and 532 nm, respectively, and 0.0076 km$^{-1}$ and 0.0064 km$^{-1}$ for extinction at 355 and 532 nm, respectively (Table 3). In addition, uncertainty is introduced as a result of the regression itself. Relative slope uncertainties range from 3.0-3.6% (Figure 3). We discuss each of these sources of error separately due to their dependence on one another. Uncertainties in both CCN concentration and HSRL-2 observations will impact uncertainty in the slope of each regression. Therefore, the assumption of each source of error being independent that is required for error propagation calculations does not hold. Rather, we present our method of deriving CCN concentration from lidar observables with such explanation of the various sources of error that will impact results.*

*In addition to observational and regression-based uncertainties, another possible source of error when applying this method stems from the specific characteristics of the data set used to develop the regression equations. The relationships analyzed in this study are specific to BBA in the SEA. Additionally, they are specific to ambient conditions with low RH (≤40-50%), S≥0.2%, and aerosol ages represented by f44 values between about 0.17-0.27. While these conditions are characteristic of the high-altitude SEA smoke plume, they will not hold in all regions and for all aerosol types. Therefore, without careful consideration of the ambient conditions and aerosol types to which the regressions derived here are applied, increased uncertainty will be introduced in lidar-derived CCN concentrations. Despite the strict conditions under which our regressions are applicable, we will explore their performance on a larger portion of the collocated data set in the following section."*

In terms of volume-averaging extinction and backscatter coefficients in our collocation method, we provide standard deviation values in the table below. These values represent the standard deviation of HSRL-2 coefficients vertically averaged that go into each collocated data point. Depending on the year, approximately 3-6 values are vertically averaged.

| | Standard Deviation of Volume-Averaged HSRL-2 Coefficients | |
|---|---|---|
| | **Absolute Mean** | **Relative Mean** |
| **BSC355** | $9.3E\text{-}05 \ km^{-1}sr^{-1}$ | 6.5% |
| **BSC532** | $4.4E\text{-}05 \ km^{-1}sr^{-1}$ | 5.9% |
| **EXT355** | $3.4E\text{-}03 \ km^{-1}$ | 3.4% |
| **EXT532** | $2.3E\text{-}03 \ km^{-1}$ | 4.6% |

We provide these values in comparison to the HSRL-2 uncertainty values given in the previous table to show that our vertical averaging of HSRL-2 coefficients into each collocated data point results in standard deviations comparable to or lower than our approximate HSRL-2 calculated uncertainties. Therefore, the volume averaging step of our collocation method does not result in averaging over highly variable backscatter and extinction coefficients.

Minor comments:

In the second line of the Figure 9 caption, ".0.5" should be replaced with "0.5"

Figure 9 was adjusted based on comments from Reviewer 1. Therefore, this issue was resolved as the caption was also changed.